# X2C: Enabling Realistic Human-to-Humanoid Facial Expression Imitation

## Abstract

The ability to imitate realistic facial expressions is essential for humanoid robots in affective human–robot communication. Achieving this requires modeling two correspondences: between human and humanoid expressions, and between humanoid expressions and robot control values. During training, we predict control values from humanoid expression images, while at execution time the control values drive the robot to reproduce expressions. Progress in this area has been limited by the lack of datasets containing diverse humanoid expressions with precise control annotations. We introduce **X2C** (Expression to Control), a large-scale dataset of 100,000 ⟨image, control value⟩ pairs, where each image depicts a humanoid robot expression annotated with 30 ground-truth control values. Building on this resource, we propose **X2CNet**, a framework that transfers expression dynamics from human to humanoid faces and learns the mapping between humanoid expressions and control values. X2CNet enables in-the-wild imitation across diverse human performers and establishes a strong baseline for this task. Real-world robot experiments validate our framework and demonstrate the potential of X2C to advance realistic human-to-humanoid facial expression imitation.

## 1 Introduction

Humanoid robots are increasingly envisioned as social agents capable of engaging humans in natural and emotionally rich interactions. A key prerequisite for such affective human–robot communication is the ability to perceive and reproduce realistic facial expressions. By imitating nuanced human expressions, humanoids can convey empathy, establish trust, and enhance interaction quality in domains such as education, healthcare, and collaborative work Stock (2016); Kanda et al. (2004); Tanaka et al. (2007); Kozima et al. (2005); Esubalew et al. (2012). Consequently, enabling humanoid robots to reproduce authentic facial expressions has become a central focus, as facial expressions play a critical role in conveying emotional cues Mehrabian (2017); Li et al. (2025).

Achieving realistic human-to-humanoid facial expression imitation is inherently challenging. It requires modeling two critical correspondences: (i) between human and humanoid expressions, which differ due to anatomical and mechanical constraints, and (ii) between humanoid expressions and robot control values, which determine the physical actuation of the robot's face. The latter constitutes an inverse problem relative to on-robot execution, where control values are applied to generate expressions. Capturing this mapping from humanoid expressions to control values is essential for constructing training pairs of the form ⟨image, control value⟩, enabling direct supervision for expression-to-control learning. Furthermore, unlike categorical emotion recognition, expression imitation requires modeling fine-grained, temporally coherent variations in facial dynamics, which poses unique challenges for learning-based methods. Moreover, affective cues are often continuous and nuanced, making them difficult to represent with coarse, categorical labels such as "happy" or "sad".

Progress in this area is hindered by the lack of datasets containing nuanced humanoid expressions specifically designed for realistic facial expression imitation. Existing resources (e.g., Smile Chen et al. (2021), Coexpression Hu et al. (2024)) are limited in scale, diversity, and annotation quality. They often include only a small number of expressions, omit asymmetric or fine-grained variations, and rely on facial landmark predictions Baltrušaitis et al. (2016), which can introduce inaccura-

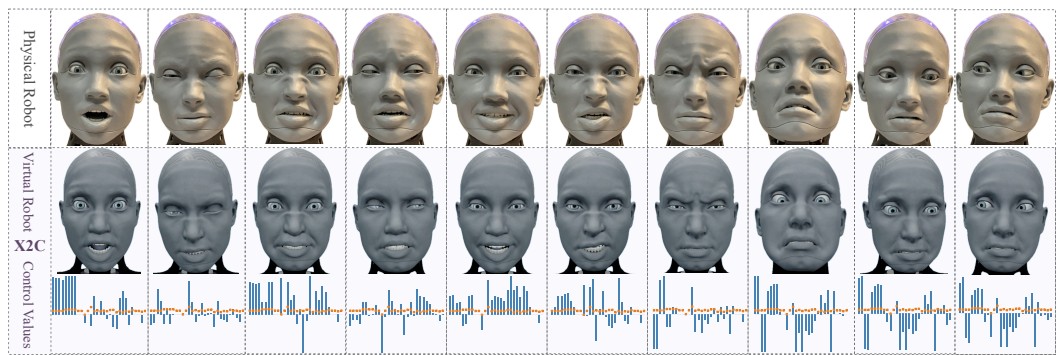

Figure 1: **Demonstration of X2C dataset examples.** Each example in the dataset consists of: (1) an image depicting the virtual robot, shown in the middle; and (2) the corresponding control values, visualized at the bottom. In these visualizations, the height of each **blue bar** represents the magnitude of the corresponding value, while the **orange dots** indicate the values in the neutral state.

Table 1: Summary statistics of existing datasets for realistic humanoid facial expression imitation.

| Dataset | Dataset Characteristics | | | | Dataset Quality | | |
|---|---|---|---|---|---|---|---|
| | Size | Asymmetric Expressions | Input Dimensionality | Annotation Dimensionality | Annotation Accuracy | Data Alignment | Data Diversity |
| **X2C** | **100,000** | ✓ | $512 \times 512 \times 3$ | **30** | ★★★★★ | ✓ | ★★★★★ |
| Smile Chen et al. (2021) | 15,000 | ✗ | $480 \times 320 \times 3$ | 10 | ★★★★ | ✓ | ★★★ |
| Coexpression Hu et al. (2024) | 1000 | ✗ | $113 \times 2$ | 11 | ★★★★ | ✓ | ★★★ |

cies. These limitations constrain the fidelity of humanoid expression imitation and impede reliable mapping from visual appearance to low-level control signals.

To address these challenges, we introduce **X2C** (Expression to Control), a large-scale dataset for realistic humanoid facial expression imitation. X2C contains 100,000 ⟨image, control value⟩ pairs, with each image depicting a humanoid robot expression annotated with 30 continuous control values representing the ground-truth actuator configuration. By providing paired visual and control data, X2C enables direct mapping from observed expressions to executable signals—a crucial step for real-world deployment—and fills a critical gap in current resources.

Building on X2C, we propose **X2CNet**, a two-stage baseline framework for human-to-humanoid expression imitation. X2CNet first transfers expression dynamics from human faces to humanoid faces in image space, preserving subtle emotional cues. It then learns the correspondence between the generated humanoid expressions and the underlying control values. This design allows the framework to handle variability across different human performers, capture fine-grained expression nuances, and produce control signals that can be directly executed on the robot. **Our main contributions are threefold:**

1. We introduce **X2C**, the first high-quality, high-diversity, large-scale dataset of nuanced humanoid facial expressions paired with precise control annotations.

2. We propose **X2CNet**, a two-stage imitation framework that establishes a strong baseline for human-to-humanoid expression learning from X2C.

3. We validate our dataset and framework through real-world demonstrations on physical humanoid robots, demonstrating the effectiveness of our approach and the potential of X2C to advance realistic humanoid expression imitation.

By bridging the gap between human expressions, humanoid expressions, and robot control, this work establishes a foundation for more natural and affective human–robot communication.

Table 2: Names of the controls and the corresponding ranges of their values.

| Jaw Pitch | Jaw Yaw | Lip Bottom Curl | Lip Bottom Depress Left | Lip Bottom Depress Middle | Lip Bottom Depress Right |
|---|---|---|---|---|---|
| $[0, 1]$ | $[0, 1]$ | $[0, 1]$ | $[0, 1]$ | $[0, 1]$ | $[0, 1]$ |
| Lip Corner Raise Left | Lip Corner Raise Right | Lip Corner Stretch Left | Lip Corner Stretch Right | Lip Top Curl | Lip Top Raise Left |
| $[0, 1]$ | $[0, 1]$ | $[0, 1]$ | $[0, 1]$ | $[0, 1]$ | $[0, 1]$ |
| Lip Top Raise Middle | Lip Top Raise Right | Nose Wrinkle | Brow Inner Left | Brow Inner Right | Brow Outer Left |
| $[0, 1]$ | $[0, 1]$ | $[0, 1]$ | $[0, 1]$ | $[0, 1]$ | $[0, 1]$ |
| Brow Outer Right | Eyelid Lower Left | Eyelid Lower Right | Eyelid Upper Left | Eyelid Upper Right | Gaze Target Phi |
| $[0, 1]$ | $[-1, 2]$ | $[-1, 2]$ | $[-1, 2]$ | $[-1, 2]$ | $[-2.3, 2.3]$ |
| Gaze Target Theta | Head Pitch | Head Roll | Head Yaw | Neck Pitch | Neck Roll |
| $[-1.1, 1.1]$ | $[-0.5, 0.3]$ | $[-0.3, 0.3]$ | $[-0.5, 0.5]$ | $[-0.3, 0.5]$ | $[-0.3, 0.3]$ |

# 2 THE X2C DATASET

X2C will be made publicly available upon acceptance of this paper, in order to preserve anonymity during the review process. It contains images of nuanced humanoid facial expressions paired with ground-truth control value annotations. A comparison of the characteristics and quality of X2C with existing datasets for realistic humanoid facial expression imitation is provided in Table 1.

In the following sections, we first present an overview of the dataset (Section 2.1) and then introduce the humanoid robot and control value preliminaries (Section 2.2). We next describe the dataset collection process, including humanoid facial expression animation curation and video recording (Section 2.3), followed by image sampling and control value computation (Section 2.4) to construct the ⟨image, control value⟩ pairs. The complete dataset collection pipeline is illustrated in Figure 4.

## 2.1 DATASET OVERVIEW

Table 3: Summary statistics of the 30 control values. The names of the controls are abbreviated using the first capital letter of each word. Full names are provided in Table 2.

| | JP | JY | LBC | LBDL | LBDM | LBDR | LCRL | LCRR | LCSL | LCSR | LTC | LTRL | LTRM | LTRR | NW |
|---|---|---|---|---|---|---|---|---|---|---|---|---|---|---|---|
| $\mu$ | 0.635 | 0.519 | 0.544 | 0.565 | 0.465 | 0.560 | 0.355 | 0.542 | 0.646 | 0.403 | 0.543 | 0.649 | 0.540 | 0.620 | 0.191 |
| $\sigma$ | 0.366 | 0.190 | 0.123 | 0.085 | 0.087 | 0.064 | 0.129 | 0.087 | 0.119 | 0.095 | 0.153 | 0.134 | 0.182 | 0.140 | 0.287 |
| $V_{\max}$ | 1.000 | 1.000 | 0.991 | 0.750 | 0.759 | 0.822 | 0.802 | 0.976 | 1.000 | 0.991 | 1.000 | 1.000 | 1.000 | 1.000 | 1.000 |
| $V_{\min}$ | 0.000 | 0.000 | 0.000 | 0.148 | 0.000 | 0.185 | 0.000 | 0.250 | 0.295 | 0.300 | 0.000 | 0.475 | 0.300 | 0.409 | 0.000 |
| $V_{\text{neu}}$ | 1.000 | 0.500 | 0.460 | 0.560 | 0.430 | 0.540 | 0.470 | 0.620 | 0.640 | 0.310 | 0.410 | 0.480 | 0.300 | 0.450 | 0.000 |

| | BIL | BIR | BOL | BOR | ELL | ELR | EUL | EUR | GTP | GTT | HP | HR | HY | NP | NR |
|---|---|---|---|---|---|---|---|---|---|---|---|---|---|---|---|
| $\mu$ | 0.605 | 0.655 | 0.613 | 0.598 | 1.110 | 1.060 | 0.989 | 0.978 | 0.074 | 0.045 | 0.004 | 0.005 | -0.002 | 0.008 | 0.002 |
| $\sigma$ | 0.236 | 0.233 | 0.211 | 0.216 | 0.521 | 0.518 | 0.311 | 0.331 | 0.311 | 0.113 | 0.036 | 0.021 | 0.048 | 0.024 | 0.009 |
| $V_{\max}$ | 1.000 | 1.000 | 1.000 | 1.000 | 2.000 | 2.000 | 2.000 | 2.000 | 2.269 | 1.082 | 0.328 | 0.300 | 0.355 | 0.204 | 0.084 |
| $V_{\min}$ | 0.000 | 0.000 | 0.000 | 0.000 | -1.000 | -1.000 | -1.000 | -1.000 | -2.269 | -0.826 | -0.371 | -0.173 | -0.413 | -0.104 | -0.158 |
| $V_{\text{neu}}$ | 0.500 | 0.500 | 0.500 | 0.500 | 1.000 | 1.000 | 1.000 | 1.000 | 0.000 | 0.000 | 0.000 | 0.000 | 0.000 | 0.000 | 0.000 |

**Diversity and Examples** Examples from X2C are shown in Figure 1, where the humanoid robot displays a broad spectrum of facial expressions. These include: (i) expressions that do not map neatly to basic emotion categories (e.g., the 4th column), (ii) variations in intensity within the same category (e.g., the last two columns, both fear), and (iii) asymmetric expressions involving specific facial units (e.g., eyelids in the 2nd column). The associated control values can directly drive the physical robot to reproduce the expressions.

The dataset encompasses expressions corresponding to basic emotions (e.g., surprise, joy, sadness Ekman (1992)) across varying intensities (e.g., fear examples differing subtly in gaze and head pose), as well as complex expressions that extend beyond canonical categories. This ensures broad **expression coverage**. Importantly, asymmetric expressions Rinn (1984) are included to simulate human-like behavior and enhance diversity. Figure 1 further illustrates correspondence between the virtual and physical robots: because they share the same control system, expressions remain consistent across embodiments.

**Characteristics and Quality** Compared to existing datasets for humanoid facial expression imitation Chen et al. (2021); Hu et al. (2024) (Table 1), X2C is larger in scale and provides higher-dimensional annotations, enabling fine-grained robot control. Unlike prior efforts that rely on tools

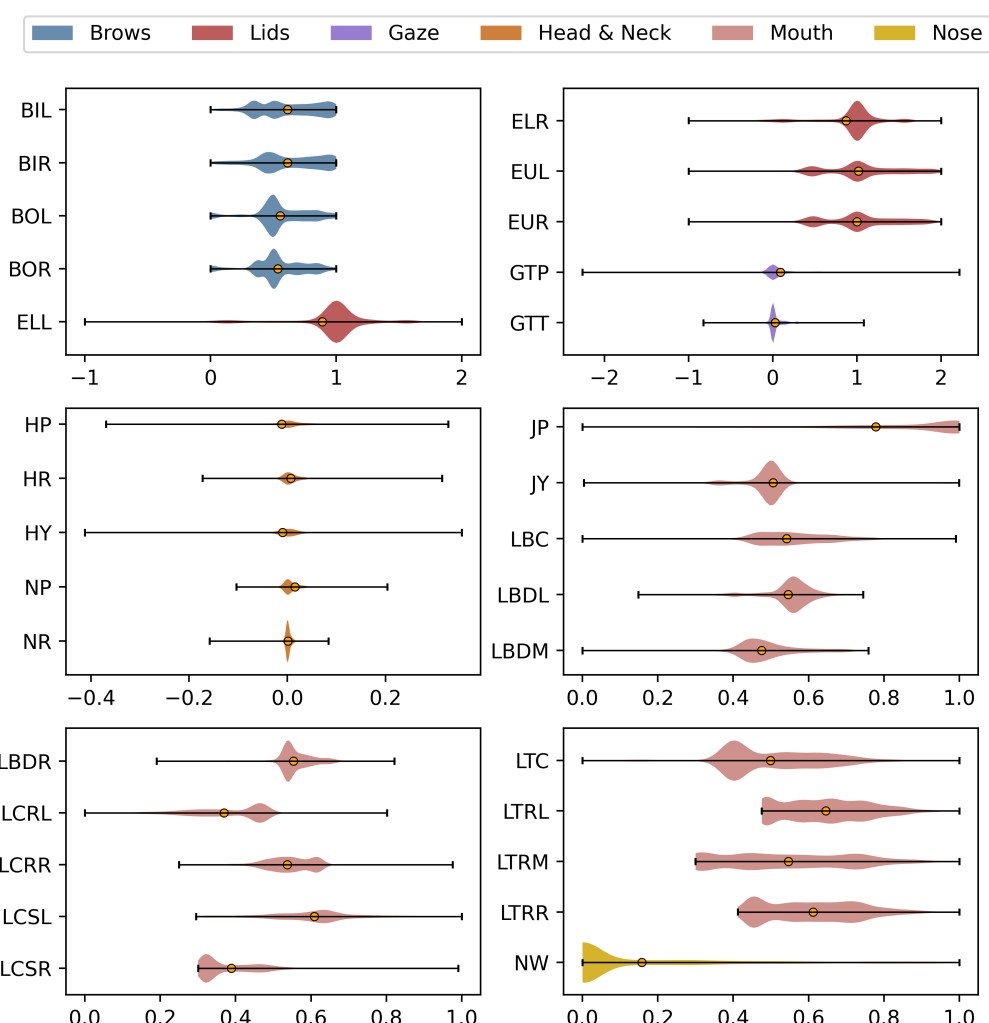

Figure 2: **Value distributions of 30 controls.** Controls for different expression-relevant units are indicated by different colors. The naming convention for controls is the same as in Table 3.

such as MediaPipe Lugaresi et al. (2019) to estimate facial landmarks (which may introduce errors), our annotations are analytically computed via interpolation functions, ensuring precise and frame-aligned control values.

A distinctive feature of X2C is its inclusion of asymmetric expressions Kowner (1995), which more closely reflect natural human behavior. This enriches the dataset and enhances the expressive capacity of humanoid robots. Table 2 lists the names and value ranges of the 30 controls explored in the dataset, while Table 3 reports their statistics, including mean ($\mu$), standard deviation ($\sigma$), minimum ($V_{\text{min}}$), maximum ($V_{\text{max}}$), and neutral state ($V_{\text{neu}}$). In many cases, $\mu$ deviates noticeably from $V_{\text{neu}}$, and several controls (e.g., JP, JY, LBC) span nearly their full achievable range, underscoring the dataset's diversity. For clarity, Figure 2 visualizes the distributions of all 30 controls.

## 2.2 THE HUMANOID ROBOT AND CONTROL VALUES

The humanoid platform used for dataset collection is Ameca, which has 32 Degrees of Freedom (DoFs) encompassing facial actuators as well as head and neck movements (Figure A5). Compared with most existing humanoid robots Zhang et al. (2025); Li et al. (2023); Faraj et al. (2021); Kerzel et al. (2017), its higher number of facial DoFs enables finer-grained and more nuanced expression control.

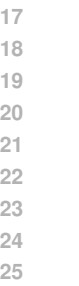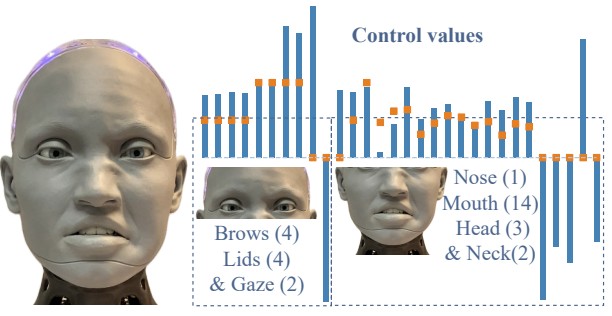

Figure 3: **An illustration of the correspondence between control values and control units.** In the control value visualization, the first 4 values control the brow movements, the next 4 control eyelid motions, and so on for the other units.

In total, 30 control values are mapped to these DoFs, driving actuators across key expression-relevant units, including the brows, lids, gaze, nose, mouth, head, and neck. The distribution of control values across these units is shown in Figure 3, while their respective ranges are provided in Table 2. Note that the number of control values does not exactly match the number of DoFs. This is because the gaze controls (Gaze Target Phi and Gaze Target Theta) drive the left and right eyes symmetrically, reflecting a human-like design since humans exhibit conjugate gaze Martinez-Conde et al. (2004).

### 2.3 HUMANOID EXPRESSION ANIMATIONS

**Environment Preparation**  Exploring the robot's expression space requires generating arbitrary combinations of control values (each sampled within its valid range) to curate a high-diversity dataset. Some combinations may produce rare expressions that most humans cannot display (Figure A2), and repeatedly driving the robot with such values can damage its silicon skin, leading to expensive repairs.

To ensure safety and minimize mechanical wear, dataset collection was conducted in a simulation environment where a **virtual counterpart** of the physical robot is available. The virtual and physical robots share the same underlying control mechanism; given identical control values, they display the same facial expressions. No sim-to-real gap Peng et al. (2018) arises here, since facial expressions are determined by facial unit movements Ekman & Friesen (1978) and disentangled from appearance (or identity) Liu et al. (2024a); Zhang et al. (2021). Images from the physical robot and its virtual counterpart are therefore equivalent in that they both encode the same expression embedding in the robot's action space, represented by control values.

Details of the data collection environment are provided in Figure A4. This environment can be accessed simultaneously by multiple certified accounts, accelerating the data collection process.

**Expression Animation Curation**  This stage involves rigging the virtual robot and creating humanoid facial expression animations. These are key-framed animations Safonova et al. (2004); Liu & Popović (2002), each defined by a sequence of critical frames that capture significant expressions at key moments, together with interpolation methods Parent (2012). The main task is to specify movements of facial control units by adjusting control values at keyframes. Several examples of key expressions are illustrated in Figure 4. Intermediate (in-between) frames are then interpolated to produce smooth animations.

The interpolation methods include **Cubic Bézier**, **Linear** and **Step** interpolations. **Cubic Bézier** interpolation produces smooth transitions by blending four control points $P_0, P_1, P_2, P_3$, where $P_0$ and $P_3$ are the start and end points, and $P_1, P_2$ are control points. The interpolation is defined over the normalized parameter $u \in [0, 1]$ Bézier (1972):

$$I(u) = (1-u)^3 P_0 + 3(1-u)^2 u P_1 + 3(1-u)u^2 P_2 + u^3 P_3, \ \ u \in [0,1]. \tag{1}$$

**Linear** interpolation creates a straight-line transition between two keyframe values $P_0$ and $P_1$ defined at times $t_0$ and $t_1$, respectively. The interpolation function is given by Foley (1996):

$$I(t) = P_0 + (P_1 - P_0) \cdot \frac{t - t_0}{t_1 - t_0}, \quad t \in [t_0, t_1]. \tag{2}$$

**Step** interpolation holds a constant value until the next keyframe. For two keyframes at times $t_0$ and $t_1$, with values $P_0$ and $P_1$, the step interpolation is defined as Alan & Mark (1992):

$$I(t) = \begin{cases} P_0, & \text{if } t \in [t_0, t_1) \\ P_1, & \text{if } t = t_1 \end{cases} \tag{3}$$

Interpolation methods and parameter choices were selected to sweep the full range of each control wherever possible, ensuring broad coverage of the expression space and promoting dataset diversity. The resulting 560 animations (1–15 seconds each) were then passed to the next processing stage.

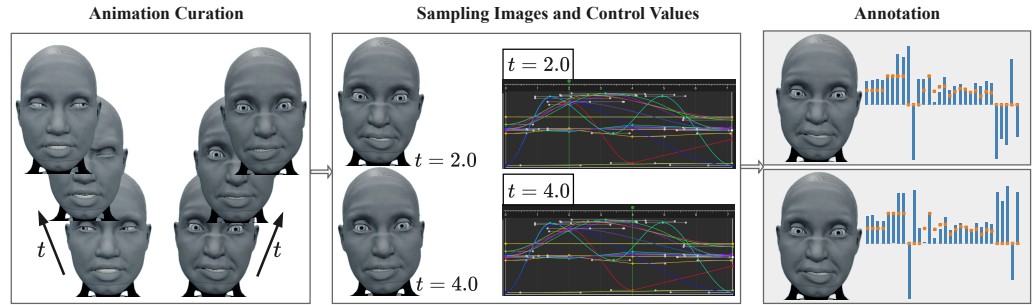

Figure 4: **Pipeline for dataset collection**. We curate humanoid facial expression animations covering all basic emotions and beyond. Images and their corresponding control values are then sampled at identical timestamps (e.g., if an image is sampled at $t = 2.0$, its control value annotation is also sampled at $t = 2.0$), ensuring temporal alignment.

### 2.4 SAMPLING AND ANNOTATION

Videos of all curated animations were recorded on the same device (MacBook Air, 2020) under consistent configurations (i.e., frame rate and robot upper-body pose) to ensure data consistency. Images were sampled at a constant timestep of $s = 0.05$ seconds and cropped to a uniform resolution of 512×512 pixels.

To ensure data uniqueness, sampled frames were processed with a structural similarity check to remove near-duplicates when similarity exceeded a threshold ($\theta = 0.99$). Given two image patches $x$ and $y$, the Structural Similarity Index (SSIM) is defined as Wang et al. (2004):

$$\text{SSIM}(x, y) = \frac{(2\mu_x\mu_y + C_1)(2\sigma_{xy} + C_2)}{(\mu_x^2 + \mu_y^2 + C_1)(\sigma_x^2 + \sigma_y^2 + C_2)}, \tag{4}$$

where $C_1 = (K_1 L)^2$, $C_2 = (K_2 L)^2$, with $L = 255$, $K_1 = 0.01$, $K_2 = 0.03$. $\mu_x$, $\mu_y$ denote the means of $x$ and $y$, $\sigma_x^2$, $\sigma_y^2$ their variances, and $\sigma_{xy}$ their covariance.

Control value annotations were obtained by retrieving the keyframes and interpolation parameters from the animation metadata, formulating the corresponding interpolation equations (Eqs. 1–3), and sampling precise control values at the same timestamps used for images. This ensures both **annotation accuracy** and **temporal alignment**.

The full dataset collection pipeline is summarized in Figure 4, which also illustrates control curve sampling at two timestamps (e.g., $t = 2.0$ and $t = 4.0$).

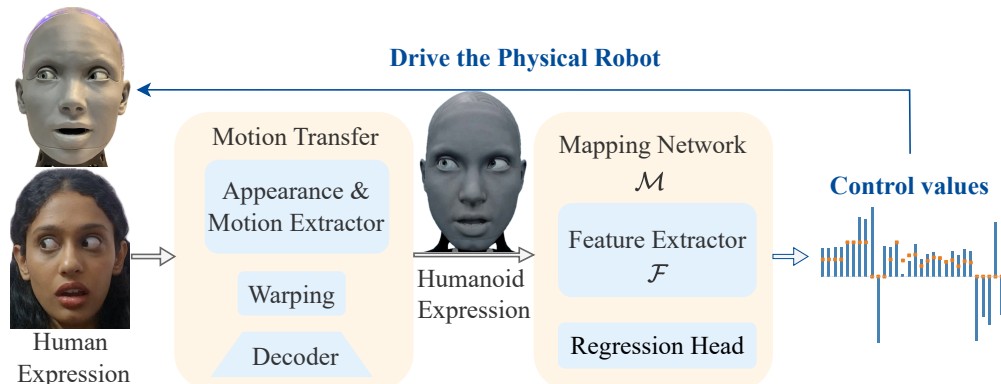

Figure 5: **Overview of X2CNet, the proposed human-to-humanoid facial expression imitation framework.** The motion transfer module captures subtle human facial expressions and combines them with the humanoid facial appearance to generate intermediate humanoid images, while the control mapping network maps these images to control values that drive the physical robot to reproduce the expressions.

## 3 THE X2CNET FRAMEWORK

### 3.1 MOTIVATION AND DESIGN

The objective of this framework is to demonstrate the value of X2C in advancing nuanced humanoid facial expression imitation. Specifically, the framework should learn directly from X2C and enable the humanoid robot to realistically mimic human facial expressions across diverse facial configurations. Designing such a system is non-trivial, as it requires bridging the gap between perceptual cues extracted from human faces and the low-level actuation commands that drive the robot.

We identify two key requirements. First, the framework must capture subtle expression dynamics from humans, including fine-grained muscle activations that differentiate nuanced emotions. Second, it must output low-level action commands that directly interface with the robot's control system Fu et al. (2024); Huang & Mutlu (2016); Spong et al. (2020). To meet the first requirement, we employ a motion transfer technique Siarohin et al. (2019b;a); Mallya et al. (2022); Guo et al. (2024); Wang et al. (2021) that warps the humanoid face according to human expressions in image space. To meet the second, we design a mapping network **trained on X2C** that learns the correspondence between humanoid expression images and their underlying control values.

X2CNet consists of two modules (Figure 5). The first is a *motion transfer module*, which generates intermediate humanoid images reflecting the input human facial motions. The second is a *control mapping module*, which predicts 30 continuous control values corresponding to expression-relevant actuation units. During inference, the motion transfer module takes human expression inputs and produces intermediate humanoid images that **resemble images in the X2C dataset** and are expected to lie close to the training distribution, while faithfully capturing the input human facial motions. This design ensures that the mapping network can reliably generate the control signals required to drive the physical robot.

For implementation, we adopt the pretrained LivePortrait Guo et al. (2024) as the motion transfer backbone, motivated by its ability to disentangle appearance and motion and its pretraining on large-scale portrait datasets Nagrani et al. (2017); Wang et al. (2020); Livingstone & Russo (2018); Liu et al. (2021). As shown in Figure 5, the module consists of an appearance extractor, a motion extractor, a warping module, and a generator. Given an input human expression, the extracted facial motions are combined with the humanoid appearance to produce a humanoid face image that resembles images in X2C.

The generated humanoid face is then passed to the mapping network, denoted by $\mathcal{M}$. This network consists of a feature extractor $\mathcal{F}$ and a regression head. $\mathcal{F}$ uses a ResNet18 backbone He et al. (2016) to capture expressive features from the humanoid images, while the regression head is a multilayer

perceptron with two hidden layers and ReLU activations Jang et al. (2022). The regression head outputs the 30 control values required to drive the robot's actuators. This two-stage design provides visual interpretability through humanoid images and direct controllability via continuous low-level actuation commands.

An alternative approach would be to directly regress humanoid control values from human facial inputs. However, such direct mapping ignores the domain gap between human and humanoid expressions and risks losing the fine-grained emotional nuances critical for realistic imitation. Our two-stage design explicitly addresses this: the motion transfer module first transfers human expression dynamics into the humanoid appearance domain, ensuring perceptual alignment, while the mapping network subsequently learns the grounded correspondence between humanoid expressions and their control parameters. This separation improves interpretability, robustness to performer variability, and generalization to in-the-wild human inputs, making X2CNet an effective and practical framework for advancing humanoid facial expression imitation.

## 3.2 EXPERIMENTS ON X2C

We compute the the mean absolute error (MAE) between the predicted and ground-truth control values, and conduct a detailed statistical analysis, including the calculation of the standard deviation (SD), standard error of the mean (SEM), and 95% Confidence Interval (CI).

We study alternative structures of the feature extractor $\mathcal{F}$ within the mapping network by replacing it with the backbone of EfficientNet-B0 (EN-B0) Tan & Le (2019), VGG16 Simonyan & Zisserman (2014) and Vision Transformer (ViT-B/16) Dosovitskiy et al. (2020). All of them are adapted from official implementations and initialized with pretrained weights.

Table 4: Comparison results based on MAE and corresponding statistical analysis.

| Method | MAE ↓ | SD ↓ | SEM ↓ | 95% CI |
|---|---|---|---|---|
| RC | 0.8951 | 0.7217 | 0.0051 | $[0.8851, 0.9051]$ |
| RT | 1.0629 | 0.8904 | 0.0063 | $[1.0505, 1.0752]$ |
| SBC | 2.3075 | 0.6250 | 0.0044 | $[2.2989, 2.3162]$ |
| LMKC | 0.1602 | 0.3402 | 0.0024 | $[0.1555, 0.1629]$ |
| **OURS** | **0.0114** | **0.0650** | **0.0005** | $[0.0105, 0.0123]$ |

Table 5: An ablation study on the feature extractor.

| $\mathcal{F}$ | MAE ↓ | SD ↓ | SEM ↓ | 95%CI |
|---|---|---|---|---|
| EN-B0 | 0.0151 | 0.0636 | 0.0004 | $[0.0142, 0.0159]$ |
| VGG16 | 0.0107 | 0.0642 | 0.0005 | $[0.0098, 0.0116]$ |
| ViT-B/16 | 0.0111 | 0.0641 | 0.0005 | $[0.0103, 0.0120]$ |
| ResNet18 | 0.0114 | 0.0650 | 0.0005 | $[0.0105, 0.0123]$ |

To validate these design choices, we conduct experiments on the X2C dataset, assessing the effectiveness of X2CNet in learning nuanced human-to-humanoid expression mappings and demonstrating its capacity to generate precise low-level control values for realistic imitation.

We split X2C into training and test sets, using 80% of them for training. We use AdamW as the optimizer with a weight decay of 0.05 and apply a *cosine schedule with warmup* as the learning rate scheduler, with an initial learning rate of 1e-3. The model is trained using the Huber loss with a threshold value of $\delta = 0.01$. Throughout training, the batch size is set to 128, and the model is trained for 100 epochs. All experiments are conducted on a single RTX 4090 GPU.

Control value prediction errors are evaluated on the test set using mean absolute error (MAE) as the performance measure. To assess the effectiveness of the mapping network, we compare our method against three baselines. The first baseline randomly samples each control value independently from a uniform distribution (RC). The second baseline randomly selects samples directly from the training set (RT) Ho & Ermon (2016); Cohen (2013). While both involve random selection, they follow different strategies. The third baseline adopts the model architecture from Liu et al. (2024b), which employs separate branches for each control value (SBC). The fourth baseline adopts the model architecture from Hu et al. (2024), predicting control values based on facial landmarks (LMKC). Quantitative results are summarized in Table 4. As shown, our method outperforms all three baselines on the test set consisting of 20,000 samples, achieving lower mean errors and smaller standard deviations. SBC shows the weakest performance because it discretizes the continuous control values into coarse labels, inherently discarding fine-grained information.

Ablation studies are conducted on the feature extractor $\mathcal{F}$ within the mapping network, The MAEs for control value prediction, along with corresponding statistical analyses are reported in Table 5.

Among the three CNN-based feature extractors—EfficientNet-B0, ResNet18, and VGG16—VGG16 achieves the best performance, followed by the transformer-based ViT-B/16. While ResNet18 performs slightly worse than both, it is significantly more lightweight and computationally efficient.

# 4 REAL-WORLD DEMONSTRATIONS

## 4.1 QUALITATIVE EVALUATION

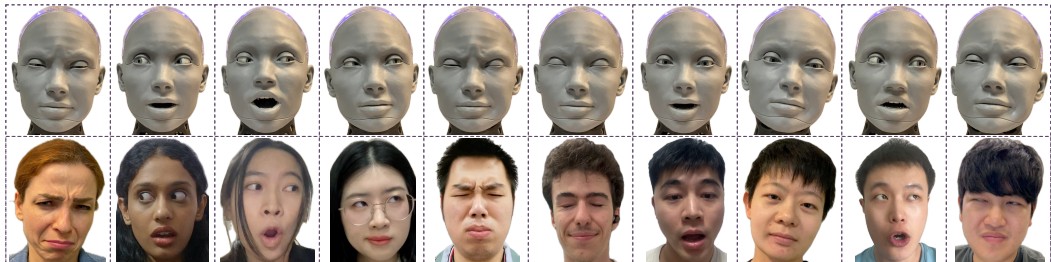

Figure 6: **Examples of realistic human-to-humanoid facial expression imitation.** Individuals express a wide range of facial expressions, with nuances reflected in features such as frown, gaze direction, eye openness, nose wrinkles, and mouth openness.

To evaluate the framework in real-world conditions, the humanoid robot must accurately replicate a wide variety of nuanced facial expressions from diverse human performers. Examples of such human-to-humanoid facial expression imitation are shown in Figure 6. The performers exhibit diverse facial configurations, including variations in facial contours, skin tones, hairstyles, and were captured under different lighting conditions. Some performers also wore accessories, such as glasses (4th) or earphones (6th). Performers were instructed to go beyond canonical expressions, incorporating subtle dynamics such as frowning (1st), gaze direction (2nd), and neck movement (8th). These expressions extend beyond standard emotion categories and can be interpreted either as blends of multiple emotions or as a single emotion expressed with varying intensities. As illustrated in Figure 6, the humanoid robot effectively replicates these expressions despite hardware constraints. An example image of the humanoid robot used in the experiments is provided in Figure A3.

These results validate the effectiveness of the human-to-humanoid imitation framework and, more importantly, demonstrate the value of X2C—our high-diversity, high-quality, large-scale dataset for humanoid learning. Additional real-world demonstrations are available on our project website: https://pi3-14159265324.github.io/X2C/.

## 4.2 QUANTITATIVE EVALUATION

To assess the perceptual realism of human-to-humanoid facial expression imitation in real-world conditions, we conduct a controlled user study comparing our proposed X2CNet against several representative baselines. The goal of this experiment is to evaluate both the absolute realism of each method's output and users' relative preference among competing approaches.

**Baselines.** We compare X2CNet with three baseline imitation models (for methods without an explicit model name in the original papers, we adopt a relevant keyword from the title):

1. **Smile**: Vision-based self-supervised imitation with a generative and inverse model Chen et al. (2021); motor commands are discretized and trained via multi-class cross-entropy.

2. **Coexpression**: Landmark-based inverse model Hu et al. (2024); predicts motor commands from normalized 2D landmarks.

3. **X2CNet-S**: Single-stage baseline predicting robot control values directly from human images, without intermediate humanoid synthesis; integrates coarse motion and fine-grained identity features. Overview in Figure A1, details in Appendix A.

**Evaluation Metrics.** To evaluate perceptual imitation quality, we conducted user studies with $m$ independent human raters. Each rater viewed paired human vs. robot expressions and rated how closely the robot expression matched the human expression on a 5-point Likert scale (1 = does not match the human expression at all, 5 = closely matches the human expression). We use the **Average User Rating (AUR)** to measure the absolute perceptual realism of each method:

$$\text{AUR} = \frac{1}{T} \sum_{t=1}^{T} r_t,$$

where $r_t$ denotes the realism score for sample $t$. Higher AUR indicates more realistic imitation. To measure *relative* perceptual preference, we use the **User Preference (User Pref.)** metric. For each of the $T$ test samples, all methods' results are shown side-by-side, and raters select the most realistic one. The proportion of selections for each method constitutes its user preference score. We report AUR and User Pref. averaged across all $m$ raters.

**Results.** Table 6 summarizes the perceptual evaluation results across all methods. The evaluation is conducted on 200 human expression test samples collected from recruited volunteers and assessed by 5 independent human raters. Across both absolute (AUR) and relative (User Pref.) metrics, the results consistently show that methods relying on coarse motion alone or landmark-only representations struggle to capture the full richness of human expressions in a physically embodied setting.

X2CNet demonstrates clear and consistent superiority across all perceptual metrics. Compared to the two prior models and our own single-stage variant, the two-stage design of X2CNet leads to substantially more realistic and interpretable robot expressions. It achieves an AUR of **4.75**, significantly higher than all baselines, indicating that participants perceive the expressions generated by X2CNet as far more lifelike. Moreover, it is selected as the most realistic method in **88%** of pairwise comparisons, demonstrating a strong overall user preference.

Table 6: Perceptual quality comparison across methods.

| Method | AUR ↑ | User Pref. (%) ↑ |
|---|---|---|
| X2CNet | **4.75** | **88** |
| Smile | 2.65 | 7.5 |
| Coexpression | 2.20 | 3.0 |
| X2CNet-S | 1.95 | 1.5 |

## 5 RELATED WORK

**Facial Expressions and Humanoid Imitation** Facial expressions are central to affective communication Mehrabian (2017); Rawal & Stock-Homburg (2022) and play a key role in human–robot interaction (HRI) De Graaf et al. (2016); Nicolescu & Mataric (2001); Ray et al. (2008); Li et al. (2025); Cao (2024). Prior work has mostly focused on analyzing human expressions, often neglecting robot facial expressiveness, which is typically limited to coarse signals such as LED indicators Barros et al. (2015); Liu et al. (2017); Johnson et al. (2013); Faria et al. (2017), restricting subtle emotional conveyance and reducing user engagement. Existing robot expression generation methods Meghdari et al. (2016); Cid et al. (2013); Ge et al. (2008); Rawal et al. (2022) often cover limited sets, while studies on nuanced humanoid imitation Antony et al. (2025); Hu et al. (2024); Chen et al. (2021); Liu et al. (2024b); Li et al. (2023) are hindered by scarce high-fidelity datasets and lack of standard benchmarks. X2C addresses this gap by providing a large-scale, diverse dataset of humanoid facial expressions with precise control annotations, enabling realistic human-to-humanoid imitation and supporting research on natural, affective HRI.

## 6 CONCLUSIONS

We present **X2C** and **X2CNet** to advance realistic human-to-humanoid facial expression imitation. Our contributions are threefold: 1) **X2C**: a large-scale, high-quality, and diverse dataset of nuanced humanoid facial expressions with precise control value annotations; 2) **X2CNet**: a novel two-stage framework for human-to-humanoid facial expression imitation; 3) Real-world demonstrations on physical humanoid robots validating the effectiveness of our approach and demonstrating the potential of X2C to enhance realistic expression imitation.

ETHICS STATEMENT

The X2C dataset contains only humanoid robot facial expression images with control value annotations and does not include any biometric data from humans. All human participants involved in the real-world robot experiments provided informed consent, and the procedures were approved by the relevant institutional review board (IRB). While some participant faces may appear in figures for illustrative purposes, care was taken to minimize identifiable information. Facial data were collected and used exclusively for research purposes. All procedures complied with relevant ethical guidelines and posed minimal risk to participants. Data from these experiments are managed responsibly to support reproducibility while upholding ethical standards for human-subject research.

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

## A APPENDIX

**LLM Usage**    LLM Usage Large Language Models (LLMs), specifically OpenAI's ChatGPT, were used as a general-purpose writing assistant. They supported grammar refinement, phrasing suggestions, and minor readability improvements of the manuscript. No part of the research design, methodology, data analysis, or results relied on LLMs, and all technical contributions were developed independently by the authors.

**Action Unit Analysis**    To further quantify expression diversity, we apply the Facial Action Coding System (FACS) Ekman & Friesen (1978), which decomposes expressions into Action Units (AUs)—elementary facial muscle movements. The presence counts of AU, detected using the Open-Face toolkit Baltrušaitis et al. (2016), are reported in Table A.1. The wide distribution of AUs across frames confirms that X2C captures both basic emotions and more complex, nuanced expressions.

| Description (AU) | Count | Description (AU) | Count | Description (AU) | Count |
|---|---|---|---|---|---|
| Inner Brow Raiser (AU01) | 38,672 | Lip Corner Puller (AU12) | 11,065 | Cheek Raiser (AU06) | 17,551 |
| Outer Brow Raiser (AU02) | 37,652 | Dimpler (AU14) | 95,297 | Lid Tightener (AU07) | 26,926 |
| Brow Lowerer (AU04) | 9,310 | Lip Corner Depressor (AU15) | 4,670 | Nose Wrinkler (AU09) | 8,427 |
| Upper Lid Raiser (AU05) | 61,602 | Chin Raiser (AU17) | 23,581 | Lips Part (AU25) | 19,516 |
| Upper Lip Raiser (AU10) | 31,231 | Lip Stretcher (AU20) | 9,969 | Jaw Drop (AU26) | 14,441 |
| Blink (AU45) | 22,908 | | | | |

Table A.1: Presence counts of Action Units (AUs) in the X2C dataset.

**Preliminary on Motion Transfer**    Here, we introduce the basic idea of implicit-keypoint–based motion transfer. Given a driving image $I_d$ that provides facial motion information and a source image $I_s$ that provides appearance information, the motion extractor estimates canonical 3D keypoints $x_{c,s}, x_{c,d} \in \mathbb{R}^{K \times 3}$, head poses $R_s, R_d \in \mathbb{R}^{3 \times 3}$, expression deformations $\delta_s, \delta_d \in \mathbb{R}^{K \times 3}$, translations $t_s, t_d \in \mathbb{R}^3$, and scale factors $s_s, s_d$ for both source and driving images. Here, $K$ denotes the number of keypoints. The keypoint transformations are formulated as:

$$\begin{cases} x_s = s_s \cdot (x_{c,s} R_s + \delta_s) + t_s, \\ x_d = s_d \cdot (x_{c,d} R_d + \delta_d) + t_d. \end{cases} \quad (5)$$

The warping network $\mathcal{W}$ generates a warping field using $x_s$ and $x_d$, which is applied to the source feature volume $f_s$. The warped feature is then passed through the generator $\mathcal{G}$ to produce an image with the driving motion and source appearance:

$$I_m = \mathcal{G}(\mathcal{W}(f_s; x_s, x_d)).$$

During inference, given a human face video stream $\{I_{d,i} \mid i = 0, \ldots, N-1\}$ and a humanoid source image $I_s$, we condition the driving facial keypoints on the source humanoid keypoints and transform them using relative motion. Specifically, we define the relative transformation of keypoints $x_{d,i}$ for each driving frame as:

$$x_{d,i} = \bar{s}_{d,i} \cdot (x_{c,s} \bar{R}_{d,i} + \bar{\delta}_{d,i}) + \bar{t}_{d,i}, \quad (6)$$

where

$$\bar{s}_{d,i} = s_s \frac{s_{d,i}}{s_{d,0}}, \quad \bar{R}_{d,i} = R_{d,i} R_{d,0}^{-1} R_s, \quad \bar{\delta}_{d,i} = \delta_s + \delta_{d,i} - \delta_{d,0}, \quad \bar{t}_{d,i} = t_s + t_{d,i} - t_{d,0},$$

denote the relative scale factor, rotation, expression deformation, and translation with respect to the source humanoid frame and the first driving human frame, respectively.

**Details of X2CNet-S**    X2CNet-S is designed to encode both coarse-grained motion cues and fine-grained appearance details of the target identity within a single unified architecture. Unlike our two-stage pipeline, X2CNet-S directly predicts robot control values without synthesizing intermediate humanoid expressions. As illustrated in Figure A1, the model consists of three main components:

- **Motion extractor**: extracts sparse implicit keypoint-based facial motion representations.
- **Dense motion network**: generates dense optical flow from the sparse motion transformation Siarohin et al. (2019b).

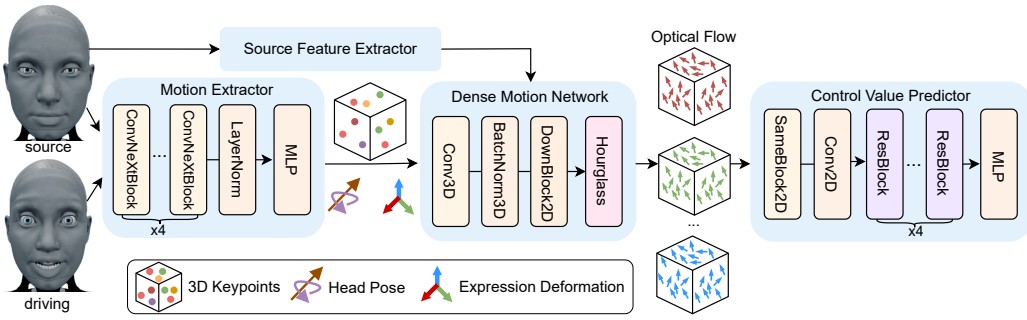

Figure A1: Overview of the proposed single-stage baseline X2CNet-S.

- **Control value predictor**: predicts robot-ready control values from the dense motion representation. To preserve fine-grained identity-specific details, appearance features extracted directly from the target face are injected into this stage.

We train X2CNet-S on the X2C dataset, which provides paired facial expressions and control values, making it well-suited for expression-to-control learning. To enhance generalization to unseen human identities, the motion extractor is initialized with pretrained weights from a large-scale human face corpus.

**Safety and Physiological Constraints** We intentionally exclude extreme values for certain controls for two reasons: (1) to avoid mechanical risks, such as excessive wear from extreme head/neck movements (HP, HR, HY, NP, NR), and (2) to maintain physiological plausibility (e.g., humans cannot fully hide their irises with gaze controls GTP, GTT, nor produce extreme lip shapes like "W" or "V"; see Figure A2).

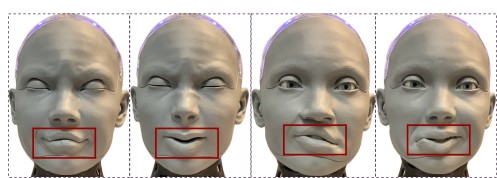

Figure A2: Examples of facial expressions that are physiologically implausible for humans.

Some combinations of control values may lead to facial expressions that are physically implausible for humans. We present several examples in Figure A2 with a focus on the mouth (('W' shape, 'V' shape, and two asymmetric cases)). Both symmetric (left two) and asymmetric (right two) cases are provided.

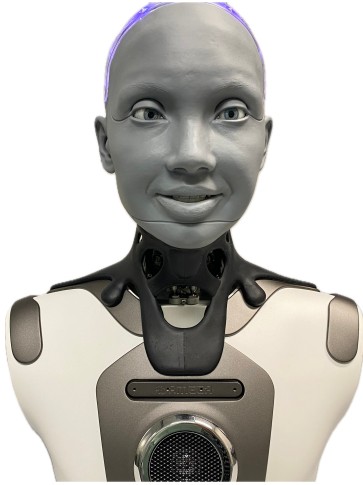

Figure A3: The humanoid robot used in real-world experiments.

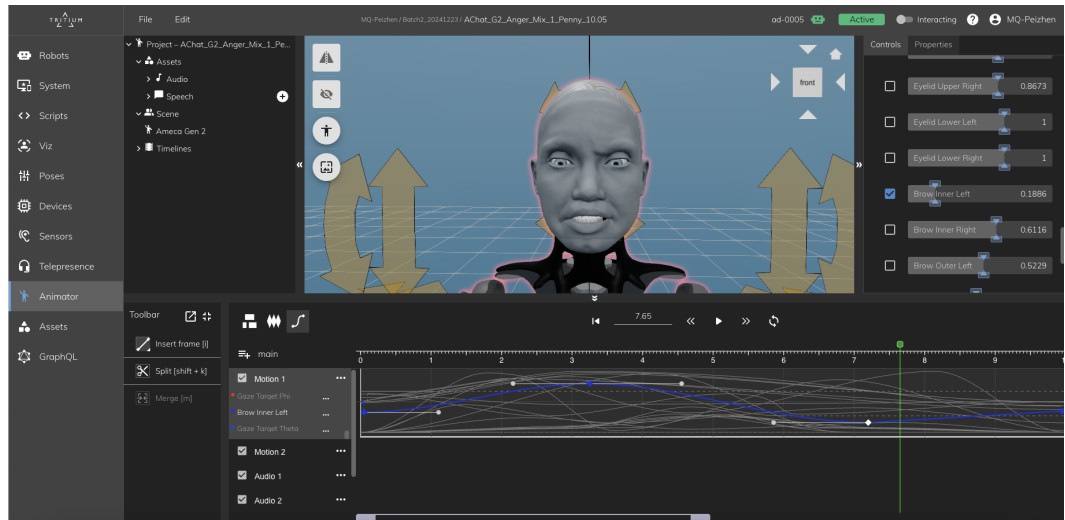

Figure A4: An overview of the environment for humanoid facial expression animation curation. A robot in the middle panel displays the desired facial expressions. It can be regarded as a shadow of the physical robot, as they share **identical** degrees of freedom (DoFs) and an expression space spanned by control values. The bottom panel shows an "activated" control (e.g., Brow Inner Left) ready to be edited, along with the corresponding value curve (i.e., the blue line) interpolated from key points (highlighted with diamond markers) at critical moments. Volunteers involved in animation curation are free to add new key points, remove or modify existing ones, and specify interpolation methods to generate intermediate values. The two panels on the left pertain to other aspects of the robot's control system.

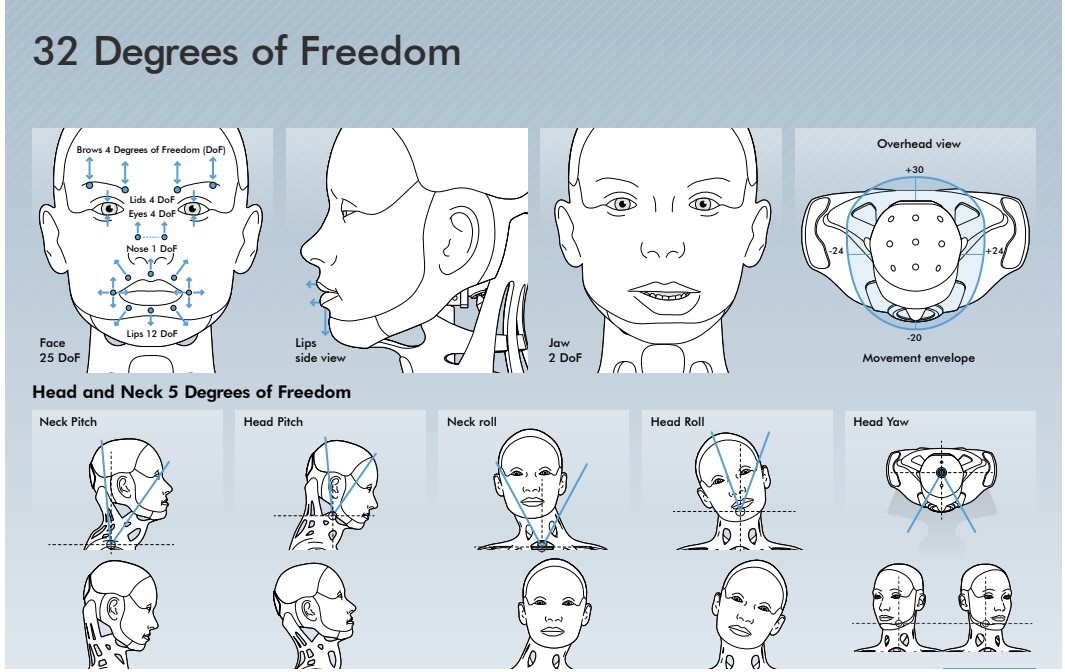

Figure A5: Demonstration of the 32 DoFs across the robot's face, head, and neck. Source: https://engineeredarts.com/robot/ameca/.

