# OpenReview forum: "X2C: Enabling Realistic Human-to-Humanoid Facial Expression Imitation"
_ICLR.cc/2026/Conference — Submitted to ICLR 2026_

### Official Review · Reviewer_jpGM · 2025-10-28

**Soundness:** 3
**Presentation:** 3
**Contribution:** 2
**Rating:** 4
**Confidence:** 3

**Summary:**

This paper introduces X2C, a large-scale dataset for realistic humanoid facial expression imitation, consisting of 100,000 image–control value pairs, and X2CNet, a two-stage framework for human-to-humanoid facial expression imitation. In the first stage, LivePortrait is employed to transfer facial motion from human faces to humanoid images; in the second stage, the model predicts 30 continuous control parameters that drive the robot’s actuators. The dataset is synthetically generated through analytical interpolation, ensuring precise annotations, and the proposed method is validated through real-world experiments on the Ameca humanoid robot.

**Strengths:**

1. High-quality dataset: The paper presents a large-scale (100K) and high-quality humanoid facial expression dataset with precisely annotated actuator control values.

2. Rigorous methodology and real-world validation: The data generation pipeline incorporates asymmetric expressions and other design strategies to ensure annotation accuracy and diversity, and the method is validated on a physical humanoid robot, demonstrating effective sim-to-real transfer and practical applicability.

3. Ethical compliance: The work exhibits solid engineering execution and provides a transparent ethical statement regarding data collection and human participation.

4. Potential impact: X2C has the potential to become a benchmark resource for research on facially expressive robots and affective human–robot interaction (HRI).

**Weaknesses:**

1. Lack of user evaluation: The study does not include user experiments assessing the perceptual realism or emotional accuracy of the reproduced expressions.

2. Engineering-oriented approach: The proposed two-stage pipeline primarily integrates existing motion transfer and regression methods, with limited algorithmic innovation.

3. Limited baselines: The paper mainly compares against a few simple or outdated methods, lacking stronger or more recent baselines, especially in terms of visual performance.

4. Dataset generalization: The experiments are conducted only on the Ameca humanoid robot, leaving it unclear whether the approach generalizes to other humanoid platforms.

5. Fine-grained expression detail imitation: The fine-grained expression details, e.g. the wrinkles could not be effectively imitated by the proposed algorithm, e.g. in Fig. 6.

**Questions:**

1. Will the generated control sequences exhibit any abrupt transitions over time?

2. Could the dataset be extended to include audio-conditioned expressions or multimodal cues (e.g., speech, gaze interaction)?

3. Beyond the reported MAE, are there any metrics for perceptual similarity of expressions?

4. Could the proposed method deal with the imitation of fine-grained expression details ?

---

> ### Author Response · Authors · 2025-11-21
> **Response to Reviewer jpGM**
>
> We sincerely thank the reviewer for their thoughtful feedback. As suggested, we have revised the paper and uploaded the updated version. We address these concerns and clarify potential misunderstandings. We have the following responses.
>
> > Lack of user evaluation: The study does not include user experiments assessing the perceptual realism or emotional accuracy of the reproduced expressions.
>
> > Beyond the reported MAE, are there any metrics for perceptual similarity of expressions?
>
>
> **Response to weakness 1 and question 3:**
> The reviewer raises two concerns: (1) the absence of user evaluation assessing perceptual realism or emotional accuracy, and (2) the lack of perceptual metrics beyond MAE. We thank the reviewer for these comments and have addressed both points by adding a comprehensive user study and new perceptual evaluation metrics.
>
> We have now included quantitative evaluation in the revised PDF (see section 4.2, Page 9). To assess the subjective and perceptual quality of the imitation, we conducted user studies in which a group of $m$ human raters watched paired expressions (human vs. robot) and rated how closely the robot expression matched the human expression on a 5-point Likert scale (1 = does not match the human expression at all, 5 = closely matches the human expression).
>
> We quantify these results using the **Average User Rating (AUR)**, defined for a single rater as the mean score across all $T$ test samples:
> $$
> \text{AUR} = \frac{1}{T} \sum_{t=1}^T r_{t},
> $$
> where $r_t$ is the realism score for sample $t$. Higher AUR values indicate that the robot's expressions more closely match the human expressions. AUR captures the absolute perceptual quality for each method.
>
> To measure relative user preference, we use the **User Preference (User Pref.)** metric. For each test sample, outputs from different methods are compared side-by-side, and raters select the robot expression that most closely matches the human expression. The proportion of selections for each method is reported as its user preference. We report mean AUR and User Pref. across all $m$ raters.
>
> The table below summarizes the results, with qualitative examples available here: [comparison-results](https://drive.google.com/file/d/1BHBdtS52IKiuge7fHbkovuOm4CSn0Go2/view?usp=sharing).
>
>
>
> | **Method**                        | **AUR** ↑ | **User Pref. (%)** ↑|
> | --------------------------------- | -------------- | ---------- |
> | X2CNet                | **4.75**           |    **88**  |
> | Smile [1]      | 2.65           |   7.5   |
> | Coexpression [2] | 2.20           | 3.0     |
> | X2CNet-S               | 1.95           | 1.5     |
>
> As shown, X2CNet achieves the highest perceptual quality, outperforming all baselines—including our additionally designed single-stage X2CNet-S (details of the model are provided in the next response)—by a large margin in both AUR and user preference. It obtains an AUR of 4.75 and is preferred in 88% of comparisons, demonstrating strong user preference for its imitation results. The evaluation is based on 200 unseen-identity human expression samples collected from volunteers and rated by 5 independent human raters.

---

> ### Author Response · Authors · 2025-11-21
>
> > Engineering-oriented approach: The proposed two-stage pipeline primarily integrates existing motion transfer and regression methods, with limited algorithmic innovation.
>
> **Response to weakness 2:** While our architecture leverages components inspired by prior work, the core contribution lies in **introducing motion-transfer–based expression imitation for human-to-humanoid control**, which— to our knowledge—has not been explored in prior literature. Existing motion-transfer methods (e.g., LivePortrait) are designed for image manipulation, not for mapping expressions to robotic actuation spaces with physical constraints. Our work establishes a novel paradigm:
> - using motion transfer to extract identity-agnostic expression dynamics, and
> - converting them into continuous robot control values that drive a physical humanoid face.
>
> This represents a new paradigm for realistic imitation and provides a seminal baseline for future work. By deploying it on a physical humanoid robot, we demonstrate the feasibility of realistic expression imitation. We encourage researchers to use the X2C dataset and our framework to explore additional possibilities.
>
> We further strengthen methodological novelty by designing **an additional single-stage baseline, X2CNet-S**. X2CNet-S is designed to incorporate both coarse-grained motion information and fine-grained facial details from the target identity within a unified architecture. It is a single-stage imitation model, meaning that—unlike the two-stage pipeline—no intermediate humanoid-expression synthesis is required. The model architecture is available here [X2CNet-S](https://drive.google.com/file/d/1n_HxfERk19pkIeEkvviJF0iLgIyFJyWf/view?usp=sharing). This model consists of three main components:
> - Motion extractor, which extracts sparse implicit keypoint-based facial motion representations;
> - Dense motion network, which generates dense optical flow from the sparse motion representation [5];
> - Control value predictor, which predicts robot-ready control values from the dense motion representation. To avoid losing fine-grained details, we directly inject feature representations extracted from the target face.
>
> We train our network on the X2C dataset, which provides paired expressions and control values and is therefore well-suited for learning control-value prediction from expressions. To ensure strong generalization to unseen human identities, we leverage a motion extractor initialized with high-quality pretrained weights obtained from large-scale human face corpora. This allows our method to benefit from broad facial variation while our training focuses on learning the expression-to-control mapping specific to the X2C setting.

---

> > ### Author Response · Authors · 2025-11-21
> >
> > > Limited baselines: The paper mainly compares against a few simple or outdated methods, lacking stronger or more recent baselines, especially in terms of visual performance.
> >
> > **Response to weakness 3:** For the mapping network, we have adopted recent baselines, including a recent study published in **Science Robotics** in 2024 [2], in our comparison experiments. While some baselines may appear simple, this does not imply they are ineffective or unsuitable for our task.
> >
> > Moreover, our work addresses the problem of realistic humanoid facial expression imitation, enabling the **physical robot** to reproduce nuanced human expressions, rather than simply reconstructing images. As this is a largely unexplored area, there are few works that achieve this level of realistic humanoid imitation, and accordingly, there are currently fewer appropriate baselines available compared to more mature fields such as computer vision.
> >
> > However, to further address the concern, we now
> >
> > 1. **Introducing alternative first-stage models.** We replace LivePortrait with two alternative motion transfer methods—one implicit-keypoint-based method (FaceVid2vid) [4], one diffusion-based method (HunyuanPortrait) [5]—and one Action Unit–guided StyleGAN-based method (AU-FEDS) [6] for comparison.
> >
> >     | **Method**      | **ACD** ↓ | **IS** ↑  |
> >     | --------------- | --------- | --------- |
> >     | LivePortrait    | 0.8883         | 1.3262        |
> >     | LivePortrait-FT  | 0.8801         | 1.3606       |
> >     | FaceVid2vid     | 0.9119         | 1.3585         |
> >     | HunyuanPortrait | 0.7950         | 1.2370         |
> >     | AU-FEDS         | 0.8867      | 1.1365         |
> >
> >
> > We curated a test dataset from the MARS dataset [7], consisting of 10,000 human facial expressions extracted across 3,105 interaction sessions covering diverse topics. This test dataset will be released along with X2C, subject to the End User License Agreement (EULA) of the original MARS dataset.
> >
> > Experimental results for the humanoid facial expression synthesis stage—evaluated using Average Content Distance (ACD) and Inception Score (IS)—are summarized in the table above. As shown, LivePortrait consistently outperforms other implicit keypoint–based methods such as FaceVid2Vid. Moreover, the fine-tuned and non–fine-tuned versions of LivePortrait yield similar performance, suggesting that the pretrained model already provides strong generalization. While diffusion-based methods achieve slightly better score in ACD, their inference speed is prohibitively slow, making them unsuitable for real-world interactive scenarios.
> >
> > 2. **Include more baseline for mapping network.** We additionally include a multi-branch control mapping method, which employ separate branches for each control value predictiono (SBC), which is published in IEEE Transactions on Robotics, 2024 [6].
> >
> >     | Method   | MAE ↓      | SD ↓       | SEM ↓      | 95% CI           |
> >     | -------- | ---------- | ---------- | ---------- | ---------------- |
> >     | RC       | 0.8951     | 0.7217     | 0.0051     | [0.8851, 0.9051] |
> >     | RT       | 1.0629     | 0.8904     | 0.0063     | [1.0505, 1.0752] |
> >     | ***SBC***      | 2.3075     | 0.6250     | 0.0044     | [2.2989, 2.3162] |
> >     | LMKC     | 0.1602     | 0.3402     | 0.0024     | [0.1555, 0.1629] |
> >     | **OURS** | **0.0114** | **0.0650** | **0.0005** | [0.0105, 0.0123] |
> >
> >
> > 3.  **Designing a single-stage baseline.** We additionally design a single-stage model that directly predicts robot control values from human facial expressions without synthesizing intermediate humanoid expressions in image space. We refer to this model as **X2CNet-S** (model details and comparison results are provided in previous responses).

---

> ### Author Response · Authors · 2025-11-21
>
> > Dataset generalization: The experiments are conducted only on the Ameca humanoid robot, leaving it unclear whether the approach generalizes to other humanoid platforms.
>
> **Response to weakness 4:** We have provided video demonstrations on **multiple** humanoid robots with different facial appearances. Please refer to our project website, with the anonymized link already included in the original paper (last line in Sec. 4, https://pi3-14159265324.github.io/X2C/). Our dataset and framework are applicable to other humanoid robots with similar or lower degrees of freedom. Since our annotations consist of control values that directly specify **how the underlying actuators should move**, analogous in spirit to Action Units in Facial Action Coding System [7], which describe expression-related muscle activations and are identity-agnostic—**rather than how the robot's face should look**—the representation is inherently generalizable.
>
> Because our task focuses on realistic humanoid imitation, the target robot platform should have a human-like face. Ameca serves as one such use case. Due to the high cost of humanoid robot platforms, we are currently unable to provide experiments on additional robots, though we are open to doing so in future work.
>
> > Fine-grained expression detail imitation: The fine-grained expression details, e.g. the wrinkles could not be effectively imitated by the proposed algorithm, e.g. in Fig. 6.
>
> **Response to weakness 5:** Actually, fine-grained expression details—such as nose wrinkles, frowns, gaze direction, and the openness of the eyes and mouth—are **successfully reproduced** on the robot's face in Fig. 6.
> For example, in the first pair (first column), the robot faithfully imitates the human's frown, and in the fifth pair, the robot reproduces the subtle nose wrinkles. To further address the reviewer’s concern, we additionally provide a clearer example of nose-wrinkle imitation here: [clearer wrinkles](https://drive.google.com/file/d/1Calacw-2ejsepYByeEUbqIWxTulKeiEj/view?usp=sharing).
>
> > Will the generated control sequences exhibit any abrupt transitions over time?
>
> **Response to question 1:** In most cases, no. As observed in the demonstration video on our project website, the expression sequences on the robot are very smooth. The primary reason for this is that we employ relative motion transformation during inference. The facial motion is adaptively adjusted to avoid abrupt transitions.
> Specifically, given a human face video stream $\\{I_{d,i} \mid i=0, \ldots, N-1\\}$ and a humanoid source image $I_s$, we condition the driving facial keypoints on the source humanoid keypoints and transform them using relative motion. The relative transformation of keypoints $x_{d,i}$ for each driving frame is defined as:
>
> $$
> x_{d,i} = \bar s_{d,i} \cdot (x_{c,s} \bar R_{d,i} + \bar \delta_{d,i}) + \bar t_{d,i},
> $$
>
> where
> $$
> \bar s_{d,i} = s_s \frac{s_{d,i}}{s_{d,0}}, \quad
> \bar R_{d,i} = R_{d,i} R_{d,0}^{-1} R_s, \quad
> \bar \delta_{d,i} = \delta_s + \delta_{d,i} - \delta_{d,0}, \quad
> \bar t_{d,i} = t_s + t_{d,i} - t_{d,0},
> $$
>
> denote the relative scale factor, rotation, expression deformation, and translation with respect to the source humanoid frame and the first driving human frame, respectively.
>
> We have also added a **Preliminary on Motion Transfer** section in the appendix of the revised PDF (Page 15).
>
> > Could the dataset be extended to include audio-conditioned expressions or multimodal cues (e.g., speech, gaze interaction)?
>
> **Response to question 2:** Yes. Although our current focus is on expression imitation from visual cues, the dataset could be extended to support audio-conditioned facial expression generation by incorporating audio or textual cues.
>
> > Could the proposed method deal with the imitation of fine-grained expression details?
>
> **Response to question 4:** Yes, this is exactly what we mean by **realistic imitation**. As shown in Fig. 6 of the paper and in the video demonstrations on our project website, fine-grained details such as frown, gaze direction, and the openness of the eyes and mouth are all successfully reproduced by the humanoid robot.
>
> [1] Boyuan Chen, Yuhang Hu, Lianfeng Li, Sara Cummings, and Hod Lipson. Smile like you mean it: Driving animatronic robotic face with learned models. In ICRA, 2021.
>
> [2] Yuhang Hu, Boyuan Chen, Jiong Lin, Yunzhe Wang, Yingke Wang, Cameron Mehlman, and HodLipson. Human-robot facial coexpression. Science Robotics, 2024.
>
> [3] Aliaksandr Siarohin, St´ephane Lathuili`ere, Sergey Tulyakov, Elisa Ricci, and Nicu Sebe. First order motion model for image animation. NeurIPS, 2019.
>
> [4] Ting-Chun Wang, Arun Mallya, and Ming-Yu Liu. One-shot free-view neural talking-head synthesis for video conferencing. In CVPR, 2021.

---

> > ### Author Response · Authors · 2025-11-21
> >
> > [5] Zunnan Xu, Zhentao Yu, Zixiang Zhou, Jun Zhou, Xiaoyu Jin, Fa-Ting Hong, Xiaozhong Ji, Junwei Zhu, Chengfei Cai, Shiyu Tang, et al. Hunyuanportrait: Implicit condition control for enhanced portrait animation. In CVPR, 2025.
> >
> > [6] Xiaofeng Liu, Rongrong Ni, Biao Yang, Siyang Song, and Angelo Cangelosi. Unlocking human-like facial expressions in humanoid robots: A novel approach for action unit driven facial expression disentangled synthesis. IEEE Transactions on Robotics, 2024.
> >
> > [7] Paul Ekman and Wallace V Friesen. Facial action coding system. Environmental Psychology & Nonverbal Behavior, 1978.

---

### Official Review · Reviewer_LVV1 · 2025-10-29

**Soundness:** 3
**Presentation:** 3
**Contribution:** 3
**Rating:** 6
**Confidence:** 4

**Summary:**

This paper introduces X2C, a large-scale dataset designed for realistic humanoid facial expression imitation, consisting of 100,000 paired samples of humanoid facial images and 30-dimensional control values. It also presents X2CNet, a two-stage framework for transferring human facial expressions to humanoid robots. The first stage transfers expression dynamics from human faces to humanoid images using motion transfer, while the second stage learns a mapping from these humanoid images to actuator control values. The dataset is collected using a simulated humanoid platform, ensuring precision and safety. The paper validates its contributions through quantitative experiments and real-world robot demonstrations, showing superior accuracy over baseline methods and demonstrating robust human-to-humanoid expression imitation.

**Strengths:**

**[Dataset Contribution]:** X2C fills an important gap in humanoid expression imitation research by providing the first large-scale, high-quality dataset of humanoid facial expressions annotated with precise control values. The dataset comprises asymmetric and fine-grained expressions, with a detailed analysis of 30 control dimensions and their distributions, as well as validation of the dataset's diversity.

**[Comprehensive experiments]:** The authors conduct both quantitative comparisons and ablation studies (Tables 4–5) showing that X2CNet significantly outperforms random and landmark-based baselines.

**[Real-world demonstrations]:** The paper provides convincing qualitative results (Figure 6, page 9) where a humanoid robot reproduces diverse and nuanced human expressions across performers, indicating successful transfer.

**[Clear presentation]:** The paper is well-written and easy to follow, with detailed methodology and good visual results.

**Weaknesses:**

**[Limited novelty in model design]:** While the dataset contribution is significant, the X2CNet architecture primarily adapts existing components (LivePortrait + ResNet18 regression), offering limited algorithmic innovation.

**[Insufficient cross-domain generalization analysis]:** The experiments mostly focus on within-dataset performance. It remains unclear how the model handles unseen human identities or lighting variations beyond those shown qualitatively. Is it basically determined by the adopted models?

**Questions:**

**[Q1]:** How well does X2CNet generalize to unseen human subjects, especially those with non-neutral identities, extreme expressions and inputs with different styles like cartoon?

**[Q2]:** Were any temporal consistency mechanisms considered to smooth control predictions over time?

---

> ### Author Response · Authors · 2025-11-21
> **Response to Reviewer LVV1**
>
> We sincerely thank the reviewer for their thoughtful feedback. In response, we have revised the paper and uploaded an updated version. We provide detailed answers to the concerns below.
>
> > [Limited novelty in model design]: While the dataset contribution is significant, the X2CNet architecture primarily adapts existing components (LivePortrait + ResNet18 regression), offering limited algorithmic innovation.
>
> **Response to weakness 1:**
> While our architecture leverages components inspired by prior work, the core contribution lies in **introducing motion-transfer–based expression imitation for human-to-humanoid control**, which— to our knowledge—has not been explored in prior literature. Existing motion-transfer methods (e.g., LivePortrait) are designed for image manipulation, not for mapping expressions to robotic actuation spaces with physical constraints. Our work establishes a novel paradigm:
> - using motion transfer to extract identity-agnostic expression dynamics, and
> - converting them into continuous robot control values that drive a physical humanoid face.
>
> This represents a new paradigm for realistic imitation and provides a seminal baseline for future work. By deploying it on a physical humanoid robot, we demonstrate the feasibility of realistic expression imitation. We encourage researchers to use the X2C dataset and our framework to explore additional possibilities.
>
> We further strengthen methodological novelty by designing **an additional single-stage baseline, X2CNet-S**. X2CNet-S is designed to incorporate both coarse-grained motion information and fine-grained facial details from the target identity within a unified architecture. It is a single-stage imitation model, meaning that—unlike the two-stage pipeline—no intermediate humanoid-expression synthesis is required. The model architecture is available here [X2CNet-S](https://drive.google.com/file/d/1n_HxfERk19pkIeEkvviJF0iLgIyFJyWf/view?usp=sharing). This model consists of three main components:
> - Motion extractor, which extracts sparse implicit keypoint-based facial motion representations;
> - Dense motion network, which generates dense optical flow from the sparse motion representation [5];
> - Control value predictor, which predicts robot-ready control values from the dense motion representation. To avoid losing fine-grained details, we directly inject feature representations extracted from the target face.
>
> We train our network on the X2C dataset, which provides paired expressions and control values and is therefore well-suited for learning control-value prediction from expressions. To ensure strong generalization to unseen human identities, we leverage a motion extractor initialized with high-quality pretrained weights obtained from large-scale human face corpora. This allows our method to benefit from broad facial variation while our training focuses on learning the expression-to-control mapping specific to the X2C setting.

---

> ### Author Response · Authors · 2025-11-21
>
> > [Insufficient cross-domain generalization analysis]: The experiments mostly focus on within-dataset performance. It remains unclear how the model handles unseen human identities or lighting variations beyond those shown qualitatively. Is it basically determined by the adopted models?
>
> > How well does X2CNet generalize to unseen human subjects, especially those with non-neutral identities, extreme expressions and inputs with different styles like cartoon?
>
> **Response to weakness 2 and question 1:**
> The reviewer asked (1) whether X2CNet generalizes to unseen identities and varying conditions, and (2) how it performs with extreme, non-neutral, or stylistically different inputs (e.g., cartoons).
> Our response below clarifies (a) how our evaluation setup explicitly tests generalization to unseen human subjects, (b) why X2CNet generalizes well due to its identity-agnostic motion representation, and (c) the intended scope and hardware limitations that make cartoon inputs and extreme expressions outside the target domain.
>
> **Generalization to Unseen Human Identities**
> To evaluate how well X2CNet handles previously unseen human subjects, we conduct user studies on **unseen-identity test samples**. A group of $m$ independent human raters viewed paired expressions (human vs.robot) and rated how closely the robot expression matched the human expression on a 5-point Likert scale (1 = does not match the human expression at all, 5 = closely matches the human expression).
>
> We quantify these results using the **Average User Rating (AUR)**, defined for a single rater as the mean score across all $T$ test samples:
> $$
> \text{AUR} = \frac{1}{T} \sum_{t=1}^T r_{t},
> $$
> where $r_t$ is the realism score for sample $t$. Higher AUR values indicate that the robot's expressions more closely match the human expressions. AUR captures the absolute perceptual quality for each method.
>
> To measure relative user preference, we use the **User Preference (User Pref.)** metric. For each test sample, outputs from different methods are compared side-by-side, and raters select the robot expression that most closely matches the human expression. The proportion of selections for each method is reported as its user preference. We report mean AUR and User Pref. across all $m$ raters.
>
> The table below summarizes the results, with qualitative examples available here: [comparison-results](https://drive.google.com/file/d/1BHBdtS52IKiuge7fHbkovuOm4CSn0Go2/view?usp=sharing).
>
>
> | **Method**                        | **AUR** ↑ | **User Pref. (%)** ↑|
> | --------------------------------- | -------------- | ---------- |
> | X2CNet                | **4.75**           |    **88**  |
> | Smile [1]      | 2.65           |   7.5   |
> | Coexpression [2] | 2.20           | 3.0     |
> | X2CNet-S               | 1.95           | 1.5     |
>
> As shown, X2CNet achieves the highest perceptual quality, outperforming all baselines—including the single-stage X2CNet-S—by a large margin in both AUR and user preference. It obtains an AUR of 4.75 and is preferred in 88% of comparisons, demonstrating strong user preference for its imitation results. The evaluation is based on 200 unseen-identity human expression samples collected from volunteers and rated by 5 independent human raters.
>
> **Why X2CNet Generalizes Well**
>
> X2CNet’s strong generalization stems from its design:
>
> - The motion-transfer stage relies on an implicit keypoint–based motion representation, which captures expression dynamics rather than identity-specific appearance.
>
> - This identity-agnostic representation naturally disentangles expression from personal facial features [3], enabling the model to generalize across diverse identities, lighting conditions, and facial variations.
> This design choice aligns with well-established findings in motion-transfer literature showing the advantage of keypoint-based implicit motion encoding for cross-identity generalization.
>
> **About Extreme Expressions and Cartoon Inputs**
> Our framework is designed specifically for **real-world human-to-humanoid imitation**, where inputs are natural human facial expressions. Therefore, domains such as cartoons fall outside our intended scope.
>
> Additionally, the physical robot platform introduces hardware limitations, including:
>
> - bounded actuator ranges,
> - material constraints of the silicone skin,
>
> which prevent the robot from fully reproducing certain extreme expressions even if recognized correctly by the model.
>
> Thus, while X2CNet generalizes well to unseen human subjects, extremely exaggerated or stylized inputs may not map faithfully onto the robot's physical morphology.

---

> ### Author Response · Authors · 2025-11-21
>
> > Were any temporal consistency mechanisms considered to smooth control predictions over time?
>
> **Response to question 2:**
> At the model level, we employ relative motion transformation during inference, where facial motion is adaptively adjusted to avoid abrupt transitions.
> Specifically, given a human face video stream $\\{ I_{d,i} \mid i = 0, \ldots, N-1 \\}$ and a humanoid source image $I_s$, we condition the driving facial keypoints on the source humanoid keypoints and transform them using relative motion. The relative transformation of keypoints $x_{d,i}$ for each driving frame is defined as:
>
> $$
> x_{d,i} = \bar s_{d,i} \cdot (x_{c,s} \bar R_{d,i} + \bar \delta_{d,i}) + \bar t_{d,i}
> $$
>
> where
>
> $$
> \bar s_{d,i} = s_s \frac{s_{d,i}}{s_{d,0}}, \quad
> \bar R_{d,i} = R_{d,i} R_{d,0}^{-1} R_s, \quad
> \bar \delta_{d,i} = \delta_s + \delta_{d,i} - \delta_{d,0}, \quad
> \bar t_{d,i} = t_s + t_{d,i} - t_{d,0},
> $$
>
> denote the relative scale factor, rotation, expression deformation, and translation with respect to the source humanoid frame and the first driving human frame, respectively.
>
> We have also added a **Preliminary on Motion Transfer** section in the appendix of the revised PDF (Page 15), which serves as an indirect smoothing approach compared to the following method.
>
> At the system level, we apply an **on-robot smoothing mechanism** during real-world execution for video- or streamed-image–based inference. For certain control units such as the head and neck, where large and rapid position changes can occur, we mitigate jitter and maintain temporal consistency by “warming up” the underlying actuators. This is achieved by interpolating between the current and target positions according to the update interval. The code snippet for head and neck control unit updates is as follows:
> ```python
> def update(self, current_time, current_position, target):
>     if self.reached:
>         self.position = target
>         self.last_updated = current_time
>         return target
>
>     if self.alive is False:
>         self.position = current_position
>         self.last_updated = current_time
>
>     t = (current_time - self.last_updated) / self.warm_time
>     if t >= 1:
>         self.reached = True
>         self.position = target
>         self.last_updated = current_time
>         return target
>
>     return lerp(self.position, target, t)
> ```
>
>
> [1] Boyuan Chen, Yuhang Hu, Lianfeng Li, Sara Cummings, and Hod Lipson. Smile like you mean it: Driving animatronic robotic face with learned models. In ICRA, 2021.
>
> [2] Yuhang Hu, Boyuan Chen, Jiong Lin, Yunzhe Wang, Yingke Wang, Cameron Mehlman, and Hod Lipson. Human-robot facial coexpression. Science Robotics, 2024.
>
> [3] Wei Zhang, Xianpeng Ji, Keyu Chen, Yu Ding, and Changjie Fan. Learning a facial expression embedding disentangled from identity. In CVPR, 2021.

---

> ### Comment · Reviewer_LVV1 · 2025-11-28
>
> I've read the feedback from the authors and I appreciate their effort in addressing my concerns.
>  I have decided to keep my score as 6

---

### Official Review · Reviewer_oFhJ · 2025-10-31

**Soundness:** 3
**Presentation:** 3
**Contribution:** 3
**Rating:** 4
**Confidence:** 4

**Summary:**

This paper introduces X2C, a large-scale dataset containing 100,000 image–control pairs for humanoid facial expression imitation, where each image depicts a humanoid robot face and is annotated with 30 continuous actuator control values. The authors additionally propose X2CNet, a two-stage imitation framework that first transfers human expressions to humanoid images using a motion-transfer model and then predicts control values with a ResNet-based regression network. The dataset is collected in simulation using key-framed animations and interpolation, ensuring precise alignment between visual appearance and control signals. Experiments compare the proposed mapping network against random baselines and a landmark-based model, showing lower mean absolute error in control prediction. Real-world robot demonstrations validate that the model-generated control values can drive a physical robot to imitate diverse human expressions.

**Strengths:**

Strengths:
- X2C provides 100k image–control pairs with precise continuous annotations. This dataset can be helpful for the broad community.
- The dataset preparation uses key-framed animation, interpolation, SSIM-based frame filtering, and control alignment with careful processing and considerations.
- The authors demonstrate the effectiveness of the method on a physical robot platform.

**Weaknesses:**

Weakness:
- The model architectures and algorithms do not have enough novel designs. I am not raising this point to question the novelty of the work itself, since I do appreciate the value of the dataset and the real-world demos, but simply not sure if the proposed method will be a good fit for ICLR. This work seems to be much more well-suited for a robotics venue.
- The primary metric is MAE over control values. There is no perceptual, behavioral, or user-study evaluation of expression quality. Including human evaluations will further strengthen the claims.
- Another suggestion is to consider adding more advanced models to explore algorithm improvements.
- Variations of the data, such as lighting changes or out-of-distribution identities, could be interesting to consider in the dataset and evaluations.
- No explicit characterization of when the model fails or limitations of the two-stage approach. Discussing a bit more on those aspects can be very helpful.
- There is a claim on "no sim-to-real gap" in the paper, but this is not carefully validated or quantified. At least some discussions on how the sim2real gap was resolved can be helpful.

**Questions:**

Please see weakness points.

---

> ### Author Response · Authors · 2025-11-21
> **Response to Reviewer oFhJ**
>
> We sincerely thank the reviewer for their thoughtful feedback. As suggested, we have revised the paper and uploaded the updated version. We address these concerns and clarify potential misunderstandings. We have the following responses.
>
> > The model architectures and algorithms do not have enough novel designs. I am not raising this point to question the novelty of the work itself, since I do appreciate the value of the dataset and the real-world demos, but simply not sure if the proposed method will be a good fit for ICLR. This work seems to be much more well-suited for a robotics venue.
>
> > The primary metric is MAE over control values. There is no perceptual, behavioral, or user-study evaluation of expression quality. Including human evaluations will further strengthen the claims.
>
> > Another suggestion is to consider adding more advanced models to explore algorithm improvements.
>
>
> **Response to weakness 1-3:**
>
> **(1) Fit to ICLR & Novelty of the Approach**
>
> First, we would like to emphasize that ICLR explicitly welcomes research at the intersection of machine learning and robotics—under the category “applications in audio, speech, robotics, ...,” which clearly includes our work (https://iclr.cc/). Numerous recent ICLR papers address robotics-focused problems [1,2,3]. For example, [1] introduces a dataset and benchmark for visual-tactile pretraining and dexterous manipulation, published at ICLR 2025. Our work similarly contributes to this line of research by focusing on fine-grained control from visual inputs, and therefore fits naturally within the scope of ICLR.
>
> We would like to emphasize that our work tackles realistic human-to-humanoid expression imitation from a novel perspective. We decouple the problem into two subproblems: first, learning the correspondence between human and humanoid expressions in image space; second, mapping humanoid expressions to robot control values. Our two-stage framework addresses these efficiently while capturing subtle, nuanced expressions that were previously unexplored.
>
> To our knowledge, we are the first to leverage motion transfer techniques, originally developed for image manipulation, for human-to-humanoid imitation. This represents a new paradigm for realistic imitation and provides a seminal baseline for future work. By deploying it on a physical humanoid robot, we demonstrate the feasibility of realistic expression imitation. We encourage researchers to use the X2C dataset and our framework to explore additional possibilities.
>
> **(2) Additional Model Design: Introducing X2CNet-S**
>
> To further demonstrate architectural novelty and explore algorithmic variants, we introduce an additional imitation model, X2CNet-S. X2CNet-S is designed to incorporate both coarse-grained motion information and fine-grained facial details from the target identity within a unified architecture. It is a one-stage imitation model—meaning that, unlike the two-stage pipeline, no intermediate humanoid-expression synthesis is required. The model architecture is available here: [X2CNet-S](https://drive.google.com/file/d/1n_HxfERk19pkIeEkvviJF0iLgIyFJyWf/view?usp=sharing). This model provides an additional architectural contribution and serves as a strong single-stage baseline. It consists of three main components:
>
> - Motion extractor, which extracts sparse implicit keypoint-based facial motion representations;
> - Dense motion network, which generates dense optical flow from the sparse motion representation [4];
> - Control value predictor, which predicts robot-ready control values from the dense motion representation. To avoid losing fine-grained details, we directly inject feature representations extracted from the target face.
>
> We train our network on the X2C dataset, which provides paired expressions and control values and is therefore well-suited for learning control-value prediction from expressions. To ensure strong generalization to unseen human identities, we leverage a motion extractor initialized with high-quality pretrained weights obtained from large-scale human face corpora. This allows our method to benefit from broad facial variation while our training focuses on learning the expression-to-control mapping specific to the X2C setting.

---

> > ### Author Response · Authors · 2025-11-21
> >
> > **(3) Perceptual and Human Evaluation**
> >
> > In response to the reviewer's suggestion, we include human perceptual evaluations to assess expression imitation quality under unseen identities.
> > A group of $m$ human raters watched paired expressions (human vs. robot) and rated how closely the robot expression matched the human expression on a 5-point Likert scale (1 = does not match the human expression at all, 5 = closely matches the human expression).
> >
> > We quantify these results using the **Average User Rating (AUR)**, defined for a single rater as the mean score across all $T$ test samples:
> > $$
> > \text{AUR} = \frac{1}{T} \sum_{t=1}^T r_{t},
> > $$
> > where $r_t$ is the realism score for sample $t$. Higher AUR values indicate that the robot's expressions more closely match the human expressions. AUR captures the absolute perceptual quality for each method.
> >
> > To measure relative user preference, we use the **User Preference (User Pref.)** metric. For each test sample, outputs from different methods are compared side-by-side, and raters select the robot expression that most closely matches the human expression. The proportion of selections for each method is reported as its user preference. We report mean AUR and User Pref. across all $m$ raters.
> >
> > The table below summarizes the results, with qualitative examples available here: [comparison-results](https://drive.google.com/file/d/1BHBdtS52IKiuge7fHbkovuOm4CSn0Go2/view?usp=sharing).
> >
> > | **Method**                        | **AUR** ↑ | **User Pref. (%)** ↑|
> > | --------------------------------- | -------------- | ---------- |
> > | X2CNet                | **4.75**           |    **88**  |
> > | Smile [8]      | 2.65           |   7.5   |
> > | Coexpression [9] | 2.20           | 3.0     |
> > | X2CNet-S               | 1.95           | 1.5     |
> >
> >
> > As shown, X2CNet achieves the highest perceptual quality, outperforming all baselines—including the single-stage X2CNet-S—by a large margin in both AUR and user preference. It obtains an AUR of 4.75 and is preferred in 88% of comparisons, demonstrating strong user preference for its imitation results. The evaluation is based on 200 unseen-identity human expression samples collected from volunteers and rated by 5 independent human raters.

---

> > > ### Author Response · Authors · 2025-11-21
> > >
> > > > Variations of the data, such as lighting changes or out-of-distribution identities, could be interesting to consider in the dataset and evaluations.
> > >
> > > **Response to weakness 4:** To clarify, our training target—the robot control values—is independent of lighting. Instead of manually recapturing the robot under multiple lighting setups (which would be costly and redundant), we can introduce lighting diversity through photometric augmentation (brightness/contrast/color jittering, shadow simulation, gamma adjustment). These augmentations provide more flexible and scalable coverage of lighting variations, while preserving the ground-truth control values.
> > >
> > > Regarding identity variations: although the robot has a fixed physical identity, this does not limit the model's generalization to unseen **human** identities. Our method operates on diverse human facial inputs during imitation, and our evaluation already includes identities that do not appear in the training set. Both the real-world imitation demonstrations and the user studies involve participants outside the training data, directly validating out-of-distribution identity generalization in practice.
> > >
> > >
> > > > No explicit characterization of when the model fails or limitations of the two-stage approach. Discussing a bit more on those aspects can be very helpful.
> > >
> > > **Response to weakness 5:** Although our two-stage framework is modular and flexible, and can effectively capture human facial dynamics, it still has several limitations.
> > >
> > > First, when human faces are only partially visible in the robot’s field of view, the system may misinterpret the input expression, leading to inaccuracies in the reproduced expression on the robot.
> > >
> > > Additionally, due to the limited Degrees-of-Freedom (DoFs) in the robot’s facial mechanism, certain expressions cannot be fully replicated. For example, the robot does not have actuators in the cheek region, so it is unable to raise or deform the cheeks as humans do. An illustration is provided here: [cheeks](https://drive.google.com/file/d/1u1GhebwXIuOmmuPOglGbtx36_ZUGXqdW/view?usp=sharing).
> > >
> > >
> > >
> > > > There is a claim on "no sim-to-real gap" in the paper, but this is not carefully validated or quantified. At least some discussions on how the sim2real gap was resolved can be helpful.
> > >
> > > **Response to weakness 6:** We included this claim in Sec. 2.3 to clarify that the virtual robot in the simulation environment can closely replicate the real physical robot. As explained in the original paper: “The virtual and physical robots share the same underlying control mechanism; given identical control values, they display the same facial expressions.” This ensures that, for our framework, the sim-to-real gap is negligible in terms of expression reproduction.
> > >
> > > [1] Qingtao Liu, Yu Cui, Zhengnan Sun, Gaofeng Li, Jiming Chen, and Qi Ye. Vtdexmanip: A dataset and benchmark for visual-tactile pretraining and dexterous manipulation with reinforcement learning. In ICLR, 2025.
> > >
> > > [2] Andrea Tirinzoni, Ahmed Touati, Jesse Farebrother, Mateusz Guzek, Anssi Kanervisto, Yingchen Xu, Alessandro Lazaric, and Matteo Pirotta. Zero-shot whole-body humanoid control via behavioral foundation models. In ICLR 2025.
> > >
> > > [3] Yan Zhao, Ruihai Wu, Zhehuan Chen, Yourong Zhang, Qingnan Fan, Kaichun Mo, and Hao Dong. Dualafford: Learning collaborative visual affordance for dual-gripper manipulation. In ICLR, 2023.
> > >
> > > [4] Aliaksandr Siarohin, St´ephane Lathuili`ere, Sergey Tulyakov, Elisa Ricci, and Nicu Sebe. First order motion model for image animation. NeurIPS, 2019.
> > >
> > > [5] Boyuan Chen, Yuhang Hu, Lianfeng Li, Sara Cummings, and Hod Lipson. Smile like you mean it: Driving animatronic robotic face with learned models. In ICRA, 2021.
> > >
> > > [6] Yuhang Hu, Boyuan Chen, Jiong Lin, Yunzhe Wang, Yingke Wang, Cameron Mehlman, and Hod Lipson. Human-robot facial coexpression. Science Robotics, 2024.
> > >
> > > [7] Xiaofeng Liu, Rongrong Ni, Biao Yang, Siyang Song, and Angelo Cangelosi. Unlocking human-like facial expressions in humanoid robots: A novel approach for action unit driven facial expression disentangled synthesis. IEEE Transactions on Robotics, 2024.

---

### Official Review · Reviewer_3r1C · 2025-11-01

**Soundness:** 3
**Presentation:** 3
**Contribution:** 3
**Rating:** 6
**Confidence:** 5

**Summary:**

This paper introduces a dataset and framework to solve the problem of humanoid robots imitating human facial expressions. The authors identify that a key challenge is the lack of data mapping a robot's appearance to its specific motor control commands.

Their solution has two parts:
- X2C Dataset: A new, large-scale dataset of 100,000 (image, control value) pairs. These are not human images; they are images of a virtual humanoid (Ameca) paired with the 30 ground-truth motor control values needed to create that exact expression.
- X2CNet Framework: A two-stage model for imitation. Stage 1 (Motion Transfer) uses a pre-trained model to "retarget" a live human's facial motion onto a static image of the humanoid. Stage 2 (Mapping Network) takes this newly generated humanoid image and, using the X2C dataset, predicts the 30 control values to send to the physical robot.

The method is validated on a physical Ameca robot, successfully imitating diverse human expressions in real-world conditions.

**Strengths:**

- The paper correctly identifies the critical bottleneck in this field: the lack of a large, high-fidelity dataset that directly links visual expressions to low-level motor commands.
- The X2C dataset is curated using a virtual robot to generate the training data, which is clever, as it bypasses the risk of physical damage and provides perfectly ground-truth control value annotations.

**Weaknesses:**

- The framework is heavily relying on the pre-trained motion transfer model (LivePortrait). Any artifact from this first stage—such as misinterpreting a subtle expression will be passed directly to the second stage. The mapping network will then correctly predict the control values for the wrong expression.
- The paper presents this as a system for affective human-robot communication, but provides no information on its computational performance (e.g., FPS or latency). It is unclear if this two-stage model can run fast enough for a truly live and natural interaction.
- There are no user study results to quantify the naturalness and imitation capability. While the authors provide strong quantitative metrics (like Mean Absolute Error) to show that their X2CNet framework can accurately predict the 30 ground-truth control values, this only measures engineering correctness. It does not measure the subjective, perceptual quality of the imitation. The paper relies entirely on qualitative evidence for this. Section 4, "REAL-WORLD DEMONSTRATIONS", and its corresponding Figure 6 are presented as the primary validation.

**Questions:**

- What is the end-to-end latency of the full X2CNet pipeline, from the camera capturing the human face to the robot's motors moving? Can the framework run at a speed (e.g., frames-per-second) that is high enough for a natural, in-the-wild social interaction?
- How do you ensure that the mapping network generalizes well from its clean training data to the (potentially noisy or artifact-filled) images produced by the LivePortrait model?

---

> ### Author Response · Authors · 2025-11-21
> **Response to Reviewer 3r1C**
>
> We sincerely thank the reviewer for their thoughtful feedback. In response, we have revised the paper and uploaded an updated version. We provide detailed answers to the concerns below.
>
> >The framework is heavily relying on the pre-trained motion transfer model (LivePortrait). Any artifact from this first stage—such as misinterpreting a subtle expression will be passed directly to the second stage. The mapping network will then correctly predict the control values for the wrong expression.
>
> **Response to weakness 1:** We acknowledge the reviewer's concern regarding potential error propagation in our two-stage imitation framework. While this issue is inherent to most two-stage pipelines, our design offers clear advantages over a one-stage approach:
> - Higher modularity and flexibility: Each stage can be optimized independently or replaced with improved models (e.g., alternative motion transfer methods).
>
> - Better interpretability: The intermediate motion-transfer outputs provide explicit visual representations, enabling error diagnosis and targeted refinement.
>
> To mitigate the effect of artifacts from the first-stage motion transfer module, we have made the following improvements:
>
> 1. **Fine-tune the motion transfer module.** We fine-tune LivePortrait on the X2C dataset (LivePortrait-FT) by formulating an image generation task following the GAN training paradigm.
>
> 2. **Introducing alternative first-stage models.** We replace LivePortrait with two alternative motion transfer methods—one implicit-keypoint-based method (FaceVid2vid) [1], one diffusion-based method (HunyuanPortrait) [2]—and one Action Unit–guided StyleGAN-based method (AU-FEDS) [3] for comparison.
>
>
>     | **Method**      | **ACD** ↓ | **IS** ↑  |
>     | --------------- | --------- | --------- |
>     | LivePortrait    | 0.8883         | 1.3262        |
>     | LivePortrait-FT  | 0.8801         | 1.3606       |
>     | FaceVid2vid     | 0.9119         | 1.3585         |
>     | HunyuanPortrait | 0.7950         | 1.2370         |
>     | AU-FEDS         | 0.8867      | 1.1365
>
>
>     We curated a test dataset from the MARS dataset [4], consisting of 10,000 human facial expressions extracted across 3,105 interaction sessions covering diverse topics. This test dataset will be released along with X2C, subject to the End User License Agreement (EULA) of the original MARS dataset.
>
>     Experimental results for the humanoid facial expression synthesis stage—evaluated using Average Content Distance (ACD) and Inception Score (IS)—are summarized in the table below. As shown, LivePortrait consistently outperforms other implicit keypoint–based methods such as FaceVid2Vid. Moreover, the fine-tuned and non–fine-tuned versions of LivePortrait yield similar performance, suggesting that the pretrained model already provides strong generalization. While diffusion-based methods achieve slightly better scores, their inference speed is prohibitively slow, making them unsuitable for real-world interactive scenarios.
>
>
>
> 3. **Designing a single-stage baseline.** We additionally design a single-stage model that directly predicts robot control values from human facial expressions without synthesizing intermediate humanoid expressions in image space. We call it **X2CNet-S**.
> X2CNet-S is designed to incorporate both coarse-grained motion information and fine-grained facial details from the target identity within a unified architecture. It is a single-stage imitation model, meaning that—unlike the two-stage pipeline—no intermediate humanoid-expression synthesis is required. The model architecture is available here: [X2CNet-S](https://drive.google.com/file/d/1n_HxfERk19pkIeEkvviJF0iLgIyFJyWf/view?usp=sharing). This model consists of three main components:
>     - Motion extractor, which extracts sparse implicit keypoint-based facial motion representations;
>     - Dense motion network, which generates dense optical flow from the sparse motion representatiotn [5];
>     - Control value predictor, which predicts robot-ready control values from the dense motion representation. To avoid losing fine-grained details, we directly inject feature representations extracted from the target face.
>
>     We train our network on the X2C dataset, which provides paired expressions and control values and is therefore well-suited for learning control-value prediction from expressions. To ensure strong generalization to unseen human identities, we leverage a motion extractor initialized with high-quality pretrained weights obtained from large-scale human face corpora. This allows our method to benefit from broad facial variation while our training focuses on learning the expression-to-control mapping specific to the X2C setting.

---

> > ### Author Response · Authors · 2025-11-21
> >
> > > The paper presents this as a system for affective human-robot communication, but provides no information on its computational performance (e.g., FPS or latency). It is unclear if this two-stage model can run fast enough for a truly live and natural interaction.
> >
> > **Response to weakness 2:** Our focus in this paper is on the proposed dataset and the novel baseline framework rather than a full affective HRI system. Therefore, we did not report system-level statistics in the initial submission. However, our X2CNet pipeline can run at around 25 FPS, with an end-to-end latency within 0.05 seconds, which is suitable for natural, in-the-wild social interaction [6]. Experimental details are provided below.
> >
> > We evaluate real-time performance using end-to-end latency, defined as the time interval between the camera capturing a human face and the robot executing the predicted control values. Latency was measured under three CPU-load conditions (idle, 50%, and 90%), induced using *stress-ng*. Each condition was repeated three times, with each run lasting 10 minutes (~10k samples). CPU utilization was monitored using *mpstat* throughout the tests. From the collected latency samples, we compute standard statistics commonly used in real-time system evaluation [7].
> >
> > - Mean latency: the arithmetic average of all latency samples, representing typical system response.
> > - P95 and P99 latency: the 95th and 99th percentiles, representing tail latency. For example, the P95 latency is the value below which 95% of the latency samples fall, capturing rare but important slow responses.
> > - Maximum latency: the largest observed latency among all samples, indicating the worst-case response.
> > - Standard deviation (jitter): the variability of latency around the mean.
> >
> > Latency evaluations were conducted on a server equipped with an Intel i9-14900K CPU (24 cores, 32 threads) and an NVIDIA RTX 4090 GPU, together with the humanoid robot running the [Tritium system](https://engineeredarts.com/software/tritium/). The results are reported below:
> >
> > | **Condition** | **Mean** | **Std** | **P95** | **P99** | **Max** |
> > | ------------- | -------- | ------- | ------- | ------- | ------- |
> > | Idle CPU      | 0.0340   | 0.0021  | 0.0384  | 0.0447  | 0.0482  |
> > | 50% Load      | 0.0413   | 0.0057  | 0.0482  | 0.0492  | 0.0518  |
> > | 90% Load      | 0.0459   | 0.0061  | 0.0557  | 0.0592  | 0.0695  |
> >
> > As shown, under idle CPU load, the mean latency is 0.0340 seconds and the P99 latency is 0.0447 seconds, indicating that the robot can provide mimicry responses within 0.05 seconds for almost all (99\%) incoming human expressions. Even under 90\% CPU load, the robot still provides mimicry responses within 0.05 seconds on average (mean latency of 0.0459 seconds). These results demonstrate the feasibility of our pipeline for natural, real-time social interaction.

---

> > > ### Author Response · Authors · 2025-11-21
> > >
> > > > There are no user study results to quantify the naturalness and imitation capability. While the authors provide strong quantitative metrics (like Mean Absolute Error) to show that their X2CNet framework can accurately predict the 30 ground-truth control values, this only measures engineering correctness. It does not measure the subjective, perceptual quality of the imitation. The paper relies entirely on qualitative evidence for this. Section 4, "REAL-WORLD DEMONSTRATIONS", and its corresponding Figure 6 are presented as the primary validation.
> > >
> > > **Response to weakness 3 and question 1:** For qualitative evaluation, we also provided video demonstrations on our project website (an anonymized link was provided in the original paper, last line of Sec. 4, https://pi3-14159265324.github.io/X2C/).
> > >
> > > We have now included quantitative evaluation in the revised PDF (see section 4.2, Page 9). To assess the subjective and perceptual quality of the imitation, we conducted user studies in which a group of $m$ human raters watched paired expressions (human vs. robot) and rated how closely the robot expression matched the human expression on a 5-point Likert scale (1 = does not match the human expression at all, 5 = closely matches the human expression).
> > >
> > > We quantify these results using the **Average User Rating (AUR)**, defined for a single rater as the mean score across all $T$ test samples:
> > > $$
> > > \text{AUR} = \frac{1}{T} \sum_{t=1}^T r_{t},
> > > $$
> > > where $r_t$ is the realism score for sample $t$. Higher AUR values indicate that the robot's expressions more closely match the human expressions. AUR captures the absolute perceptual quality for each method.
> > >
> > > To measure relative user preference, we use the **User Preference (User Pref.)** metric. For each test sample, outputs from different methods are compared side-by-side, and raters select the robot expression that most closely matches the human expression. The proportion of selections for each method is reported as its user preference. We report mean AUR and User Pref. across all $m$ raters.
> > >
> > > The table below summarizes the results, with qualitative examples available here: [comparison-results](https://drive.google.com/file/d/1BHBdtS52IKiuge7fHbkovuOm4CSn0Go2/view?usp=sharing).
> > >
> > >
> > > | **Method**                        | **AUR** ↑ | **User Pref. (%)** ↑|
> > > | --------------------------------- | -------------- | ---------- |
> > > | X2CNet                | **4.75**           |    **88**  |
> > > | Smile [8]      | 2.65           |   7.5   |
> > > | Coexpression [9] | 2.20           | 3.0     |
> > > | X2CNet-S               | 1.95           | 1.5     |
> > >
> > > As shown, X2CNet achieves the highest perceptual quality, outperforming all baselines—including the single-stage X2CNet-S—by a large margin in both AUR and user preference. It obtains an AUR of 4.75 and is preferred in 88% of comparisons, demonstrating strong user preference for its imitation results. The evaluation is based on 200 unseen-identity human expression samples collected from volunteers and rated by 5 independent human raters.

---

> ### Author Response · Authors · 2025-11-21
>
> > How do you ensure that the mapping network generalizes well from its clean training data to the (potentially noisy or artifact-filled) images produced by the LivePortrait model?
>
> **Response to question 2:**
> To improve the generalizability of the mapping network and enhance its robustness to potentially noisy or artifact-filled inputs from the LivePortrait model, we adopt a **feature alignment** training strategy, which exposes the model to inputs from multiple sources (the original training set and generated images simulating inference inputs) and encourages their features to be consistent.
> Specifically, for each image $I_x$ in X2C, we generate a corresponding transformed image $\hat{I}_x$ using the motion transfer module:
>
> $$
> \hat{I}_x = \mathcal{T}(I_x)
> $$
>
> During training, we compute the feature for the original image:
>
> $$
> \phi(I_x) = \mathcal{F}(I_x)
> $$
>
> and also extract the feature for the transformed image:
>
> $$
> \phi^\star(\hat{I}_x) = \mathcal{F}^\star(\hat{I}_x)
> $$
>
> without tracking gradients. Here, $\phi^\star(\hat{I}_x)$ acts as a target feature from a simulated inference input, and the superscript $\star$ denotes features that do not require gradients. This ensures that both features are produced by the same network with shared weights, while preventing the alignment step from interfering with control value prediction.
>
> The feature alignment loss $\mathcal{L}_{\text{align}} $ is defined as the mean squared error (MSE) between features.
>
>
> The overall training objective is then:
>
> $$
> \mathcal L_{\text{total}} = \mathcal L_{\text{ctrl}} + \lambda_\text{align} \mathcal{L}_\text{align}
> $$
>
> where $\lambda_{\text{align}}$ is a weighting factor and $\mathcal{L}_{\text{ctrl}}$ is the Huber loss for control value regression.
>
> To evaluate the impact of the feature alignment (f.a.), we conduct ablation experiments on the test samples. Qualitative results are available here: [ablation-feature-alignment](https://drive.google.com/file/d/1zX7OyFU-8GMOHUcMCyWCb_eQ9liEZE3p/view?usp=sharing). As shown, with feature alignment, subtle details—such as the openness of the mouth and eyes—align more closely between the robot and the human.
>
>
>
>
> [1] Ting-Chun Wang, Arun Mallya, and Ming-Yu Liu. One-shot free-view neural talking-head synthesis for video conferencing. In CVPR, 2021.
>
> [2] Zunnan Xu, Zhentao Yu, Zixiang Zhou, Jun Zhou, Xiaoyu Jin, Fa-Ting Hong, Xiaozhong Ji, Junwei Zhu, Chengfei Cai, Shiyu Tang, et al. Hunyuanportrait: Implicit condition control for enhanced portrait animation. In CVPR, 2025.
>
> [3] Xiaofeng Liu, Rongrong Ni, Biao Yang, Siyang Song, and Angelo Cangelosi. Unlocking human-like facial expressions in humanoid robots: A novel approach for action unit driven facial expression disentangled synthesis. IEEE Transactions on Robotics, 2024.
>
>
> [4] Siyang Song, Micol Spitale, Xiangyu Kong, Hengde Zhu, Cheng Luo, Cristina Palmero, German Barquero, Sergio Escalera, Michel Valstar, Mohamed Daoudi, et al. React 2025: the third multiple appropriate facial reaction generation challenge. In MM, 2025.
>
> [5] Aliaksandr Siarohin, St´ephane Lathuili`ere, Sergey Tulyakov, Elisa Ricci, and Nicu Sebe. First order motion model for image animation. NeurIPS, 2019.
>
> [6] Valentin Forch, Thomas Franke, Nadine Rauh, and Josef F Krems. Are 100 ms fast enough? characterizing latency perception thresholds in mouse-based interaction. In EPCE, 2017.
>
> [7] Jeffrey Dean and Luiz Andr´e Barroso. The tail at scale. Communications of the ACM, 2013.
>
> [8] Boyuan Chen, Yuhang Hu, Lianfeng Li, Sara Cummings, and Hod Lipson. Smile like you mean it: Driving animatronic robotic face with learned models. In ICRA, 2021.
>
> [9] Yuhang Hu, Boyuan Chen, Jiong Lin, Yunzhe Wang, Yingke Wang, Cameron Mehlman, and Hod Lipson. Human-robot facial coexpression. Science Robotics, 2024.

---

### Meta-Review · Area_Chair_8Fgf · 2026-01-03

**Summary:**

While reviewers acknowledge that the dataset itself is valuable, the consensus is that the dataset constitutes the primary and only substantial contribution of the paper, with the accompanying model serving mainly as a baseline. Several reviewers raised concerns about generalization and evaluation. In particular, Reviewer LVV1 and Reviewer jpGM questioned whether the approach generalizes beyond the single humanoid platform, and experiments are largely confined to one robot embodiment and that cross-platform or cross-domain generalization is not convincingly demonstrated. Reviewer 3r1C and Reviewer jpGM emphasized the initial lack of user studies and perceptual evaluation, arguing that MAE over control values measures engineering correctness but does not adequately capture expressive quality, which is also an especially critical issue for affective human–robot interaction.

Beyond these points, reviewers also raised concerns about limited algorithmic novelty, as the proposed framework primarily integrates existing motion-transfer models with standard regression networks, making the method engineering-driven rather than conceptually novel, and potentially better suited to a robotics-focused venue than a general ML conference. Additional weaknesses include strong dependence on a pretrained first-stage motion-transfer model, raising questions about error propagation and robustness; It is unclear what are failure modes and limitations, particularly for extreme expressions or fine-grained facial details.

**Reviewer Concerns:**

**Concerns addressed by the rebuttal:**

The rebuttal makes a genuine effort to respond to several reviewer concerns, particularly around evaluation completeness. The authors added user studies to assess perceptual quality and imitation fidelity, reported latency and real-time performance, and introduced additional ablations and alternative baselines, including a single-stage variant and alternative motion-transfer backbones. These additions help clarify that the system can run in real time and that the proposed pipeline performs well under the authors’ chosen evaluation settings. One reviewer acknowledged these additions and appreciated the extra empirical effort.

**Concerns that remain outstanding:**

Despite these improvements, several core concerns raised by the reviewers remain unresolved. Most importantly, the paper’s primary contribution is still the dataset, while the accompanying method is largely viewed as an engineering integration of existing components (e.g., pretrained motion transfer and regression networks), with limited algorithmic novelty. Concerns raised by Reviewer LVV1 and Reviewer jpGM regarding **generalization** have not been fully addressed: experiments are still heavily centered on a single humanoid platform, and claims of cross-domain or cross-robot generalization are not convincingly demonstrated.

In addition, although user studies were added, some reviewers questioned whether the evaluation setup is sufficient to establish broad perceptual validity, given the relatively small scale and controlled conditions. The strong dependence on a pretrained first-stage motion-transfer model also remains a concern, as error propagation and robustness are not fully justified. It would be better to have quantative evaluation. Overall, while the rebuttal strengthens the paper empirically, it does not fundamentally change the assessment that the contribution is dataset-centric, with limited methodological innovation and incomplete generalization evidence, which informed the decision to reject.

**Reviewer Scores:**

During the discussion phase, one reviewer provided additional feedback but ultimately chose to keep their score at 6, without further increase. Given the mixed initial scores, the Area Chair carefully examined the key concerns raised across reviews. In particular, issues around dataset generalization, extension to additional robot platforms, and the incremental nature of the technical contributions place this work in a borderline category. While the rebuttal adds clarifications and additional experiments, the paper would benefit from quantitative evidence demonstrating how error propagation in the two-stage pipeline is mitigated by the proposed design, which would make the central claims more convincing. Given the limited acceptance capacity, and considering that the contribution does not clearly stand out relative to competing submissions, the decision is to reject.

---

### Decision · Program_Chairs · 2026-01-26

Reject